# Buckling Analysis and Optimization of Stiffened Variable Angle Tow Laminates with a Cutout Considering Manufacturing Constraints

Wei Zhao [1,*] , Rakesh K. Kapania [2]

1   School of Mechanical and Aerospace Engineering, Oklahoma State University, Stillwater, OK 74078, USA
2   Kevin T. Crofton Department of Aerospace and Ocean Engineering, Virginia Polytechnic Institute and State University, Blacksburg, VA 24061, USA; rkapania@vt.edu
*   Correspondence: wzhao@okstate.edu; Tel.: +1-405-744-6446

**Abstract:** Variable angle tow laminates (VAT) and stiffeners are known to redistribute the in-plane load distribution and tailor the buckling mode shapes, respectively, for improving structural performance. To leverage the benefits of using VAT laminates in the practical applications, in the present paper, we discuss buckling load maximization conducted for a stiffened VAT laminated plate with a central cutout considering VAT laminate manufacturing constraints. Three representative boundary conditions as seen in the aerospace structures are considered: in-plane axial displacement, in-plane pure shear, and in-plane pure bending displacements. Two common manufacturing constraints, the one on the automatic fiber placement (AFP) manufacturing head turning radius and the other on the tow gap/overlap, while fabricating VAT laminates are considered in the laminate design. These two manufacturing constraints are modeled by controlling the fiber path radius of curvature and tape parallelism in optimizing the fiber path orientations for the VAT laminates. Stiffener layout and fiber path angle for the VAT laminated plates are both considered in the buckling load maximization study. To avoid using a fine mesh in modeling the stiffened VAT laminates with a cutout when employing the finite element analysis during the optimization, the VAT laminated plate and the stiffeners are modeled independently. The displacement compatibility is enforced at the stiffener–plate interfaces to ensure that the stiffeners move with the plate. Particle swarm optimization is used as the optimization algorithm for the buckling load maximization study. Optimization results show that, without considering AFP manufacturing constraints, the VAT laminates can increase the buckling loads by 21.2% and 12.4%, respectively, comparing to the commonly used quasi-isotropic laminates and traditionally straight fiber path laminates for the structure under the in-plane axial displacement case, 19.7% and 12.5%, respectively, for the in-plane shear displacement case, and 62.1% and 26.6%, respectively, for the in-plane bending displacement case. The AFP manufacturing constraints are found to have different impacts on the buckling responses for the VAT laminates, which cause the maximum buckling load to be 9.3–10.1%, 3.0–3.2%, and 23.2–29.8% less than those obtained without considering AFP manufacturing constraints, respectively, for the present studied model under in-plane axial, shear, and bending displacements.

**Keywords:** variable angle tow laminate; stiffened laminated plate; buckling analysis; finite element method; optimization

## 1. Introduction

Lightweight structure has been considered as one important index in the structural design for airplane, launch vehicles, spacecraft exploration vehicles, and space habitats [1]. Cutouts are widely used in these vehicles for windows, doors, riveted joints, etc. These imperfections, as used in the primary structure, require structural enhancements, such as using a thicker geometry or a stronger material, to avoid stress-/strain-concentration-induced failure, which results in a heavier design. The advancement in the automated fiber

placement manufacturing for variable angle tow laminates [2] and the co-curing process [3] or the pultruded rod stitched efficient unitized structure concept [4] for the integrally stiffened panels enlarge the design space in structural design for these applications. This allows one to design a highly tailored lightweight structure while satisfying both the stiffness and strength requirements.

As compared to the traditional laminates with straight fiber path in each layer, variable angle tow (VAT) laminated structure improves the design space in fiber ply orientations for each layer, which can further improve the structural performance. Hyer and Charette [5] studied the buckling resistance and tensile capacity of a plate with a central cutout by optimizing fiber paths for the plate. The plate was divided into several segments where each segment has its own fiber path orientations. The optimal straight fiber paths for an infinite number of segments is equivalent to using curvilinear fiber path for composite structures. The study about the improvement in the buckling and tensile capacities first demonstrated the potential of using curvilinear fiber path for a further improvement on structural performance as compared to using the straight path fiber over the whole region of the plate.

Gürdal and Olmedo [6] derived exact closed-form solutions for the displacement and stress resultants for the variable stiffness panels under in-plane displacement boundary conditions. They found that changing the elastic properties using VAT laminates will lead to stress gradient among structures and concluded that the VAT laminates can tailor the stress distribution to improve the structural performance. Wu et al. [7,8] conducted both numerical and experimental studies on VAT laminated plates under compressive loads and observed an increased load-carrying capability of the VAT laminated structures comparing to the traditional straight-fiber laminates design.

Lopes et al. [9] conducted progressive damage and failure analyses for the VAT laminated panels with a central cutout under in-plane axial displacements. Research studies show that curvilinear-fiber laminated plates have up to 56% higher strength than straight-fiber laminates, which greatly postpones the damage initiation. Unlike the traditional stress concentration induced failure near the hole's edge, for the composite panels fabricated using the tow-steered laminates, the central hole is no longer the main geometrical cause for failure of a notched panel [9]. This is because of the load redistribution capacity of a VAT laminated panel that increases its buckling load and delays the initiation of damage and final structural failure. Lopes et al. [10] also showed that it is possible to design and manufacture composite panels whose failure responses are insensitive to the existence of a central hole. This cutout insensitivity does not involve an increase in the structural mass but only involves the steering of the fibers in the plane of the laminate, through redistributing the loading toward the plate supported edges.

Wu et al. [11,12] studied the structural performance of composite tow-steered shell with cutouts both numerically and experimentally. Both numerical and test results showed that using tow-steered laminates can increase the load capacity when designing structures with cutouts. Khania et al. [13] experimentally studied the failure loads, failure modes, and weights of the tested panels using a quasi-isotropic laminate, constant stiffness, and variable stiffness laminates. A pure tension load was applied in all the tested panels. The results indicate that the variable stiffness laminate is capable of sustaining significantly larger loads, before failure, than the constant stiffness and quasi-isotropic laminates of equal weight. In addition, Zucco et al. [14] studied the static and buckling responses for a representative wingbox skin panel with an elliptical hole. The use of continuous tow steering laminates increases the panel buckling load by 26% and also increases the panel weight by 17% compared to those of using straight fiber path laminates. In addition, the use of continuous tow laminates reduces the maximum direct strain that occurs near the hole's edge by 27% under the in-plane tensile load condition. Hao et al. [15] studied buckling optimization of unstiffened composite panels with cutouts using variable angle tow laminates using an isogeometric method for structural analysis. Linear variation function is used to describe the curvilinear fiber path of the composite laminates. The

optimized design shows that the VAT laminates can be used to improve the plate's buckling responses.

A drawback of VAT laminates is that many defects could be generated in them during the AFP manufacturing process, such as fiber tape wrinkling, gaps and/or overlaps, and etc., which would influence both the VAT laminates panel stiffness and mass properties. Without considering the AFP manufacturing constraints on fabricating VAT laminates, the benefits of using VAT laminates for improved structural performance cannot be leveraged for practical applications.

Gürdal et al. [2] reviewed three different fabrication issues encountered during AFP manufacturing for VA laminates: curvature constraint; gap/overlap and interweaving of plies with thickness variation. Constant curvature arcs were considered for describing the curvilinear tow paths while satisfying the head turning radius constraint of the AFP machines. Heinecke and Willberg [16] reviewed different AFP manufacturing induced imperfections in composite parts and pointed out that the gaps and overlaps are predominant among all identified defect classes. For the gaps generated during the AFP manufacturing for the VAT laminates, the resin is typically used to fill those gap areas. There are two common manufacturing methods: tow-drop and overlap, which are used to completely cover the entire structure during the VAT laminates fabrication. The tow-drop method requires the tow-placement machine to cut tows individually to prevent the overlap regions during the AFP manufacturing process [9]. The tow-drop method results in constant thickness panels that contain small wedge-like areas free of fibres due to the dropping of the individual, finite thickness tows. The overlap method would cause a nonuniform thickness distribution of laminates. To avoid overlaps, Gürdal et al. [2] used a shift method to shift a new tape, identical in the shape to the reference path, which is laid down next to the reference tape with a distance perpendicular to the variation direction.

Some manufacturing constraints have been considered in the VAT laminates structural and design optimization studies. Among them, the AFP head turning radius constraint and the gap/overlap have received significant interest. The minimal AFP machine head turning radius has been converted into the controlling fiber path curve radius of curvature, which can be considered through parameterization of the fiber path angles for the VAT laminates [17–20]. Blom et al. [21] modeled the tow-top areas in the finite element model of the VAT laminates. An extremely fine mesh was developed to capture the gaps precisely. Each node of the finite element model is checked whether it is in a tow-drop area or in a fiber region. If the point is found in the tow-drop area, this element has material properties of resin matrix, not those of the fiber-reinforced laminates. To quantify the thickness variation due to the gap/overlap for a built-in fabricated VAT laminated panel, Fayazbakhah et al. [22] used a defect layer method to capture the gaps and overlaps in the AFP fabricated VAT laminates. This approach introduces embedded defects to a regular composite material using modified material properties or thickness. The defect area percentage is used to modify the properties or the thickness of this regular composite layer. A fine mesh was used to capture the defects as accurately as possible. It was found that the gaps in the VAT laminates reduce the buckling load improvement by 15% as compared to the laminates where gaps are ignored. Nik et al. [23] considered the impact of the gap/overlap of the VAT laminates fabricated using the AFP manufacturing on maximizing the in-plane stiffness and the buckling load of the VAT laminates. They observed that the largest number of tows with the smallest width yields the minimum gap/overlap areas percentages within the laminate. Marouene et al. [24] studied the compressive strength of a VAT laminated panel with a central hole considering the modeling on the gap and overlap areas as seen in the VAT laminates. Lazano et al. [25] developed a tool that improves the manufacturability of fiber-steered laminates by controlling gaps, overlaps, and path curvatures using the built-in geometry tools available in a computer-aided design software, CATIA. Tooren et al. [26] studied stiffness corrections for overlaps and gaps in steered composite panel optimization. A density functional method is used to translate the effect

of discrete gaps and overlaps into a continuous correction of the reduced stiffness matrix of the unstiffened VAT laminates.

For defect-free VAT laminates during the design optimization studies for manufacturable VAT laminates, Brooks and Martins [18] developed a family of gap/overlap-free and curvature-free tow steered patterns for the VAT laminates using relationship between curl and divergence for the rotated tow patterns. Peeters et al. [27] imposed steering constraints on the norm of the gradient of the prescribed fiber path to ensure their smoothness during the stacking sequence optimization of a VAT laminated plate with a central hole. These approaches are used to control the fiber path of the VAT laminates to avoid the potential gap/overlap generated in the VAT laminates manufacturing.

The use of VAT laminates for structural design may introduce large stresses near the plate's supported edges and possibly results in material failures near the support. This is because the load redistribution in a VAT laminate is such that the maximum stress resultants are moved from the buckling mode peak to the structure support edges. As an alternative approach for improving the structural bucking performance, stiffeners are both more practical and efficient to increase the buckling load by tailoring the buckling mode shape wavelength. Buckling analysis and optimization results showed that the variable stiffness designs are promising for lightweight structural designs where the buckling constraint is active in the design studies [20,28,29].

To explore the maximum benefit of using VAT laminates for improving buckling performance of stiffened plates with cutouts under various loading conditions, a finite element method is employed in the present study. However, the modeling of the stiffened VAT laminates with cutouts would be time-consuming when a design also considers to optimize the stiffeners' shape and layout. The change of the stiffeners shape and layout requires a remeshing of the whole finite element model, resulting in a repeated meshing of the stiffened plate with cutouts during design studies. The automatically generated finite element mesh during design optimization studies is required to capture stress/strain concentration near the hole edges. A previously developed tool for the stiffener shape optimization, EBF3PanelOpt [30], used an extremely fine mesh for the whole structure to avoid excessive meshing failures, which results in a computational expensive structural analysis. In this study, we employ a different modeling approach, which was developed in our previous works toward studying the buckling responses of stiffened plate with curvilinear stiffeners [31,32] and stiffened VAT laminated plates [20]. This modeling approach allows us to develop independent elements for the plate and the stiffeners. The converged mesh for the plate with cutouts can be kept same with the stiffener shape/layout during the shape optimization. For brevity, this method will not be explained in detail. However, a reader can find the description in references [31,32].

In the present work, we study the fiber path orientation for VAT laminates toward improving the buckling performance of stiffened plates with cutouts. Three representative loading conditions are considered: in-plane axial displacement, in-plane pure shear displacement, and in-plane pure bending displacement. These three boundary conditions are commonly seen in the fuselage panels, wing skin panels, rib panels, and spar panels as used in the aerospace structures. For a manufacturable design of stiffened VAT laminated plate with cutouts, straight stiffeners and two common AFP manufacturing constraints on fabricating VAT laminates are considered. Furthermore, to the best of the authors' knowledge, there is no available research on buckling analysis and optimization of stiffened VAT laminates with cutouts under various boundary conditions. The present work can provide guides and benchmarks toward using VAT laminates in designing such aerospace structures under different dominant loading conditions as commonly seen in the aircraft design.

The paper is organized as follows. In Section 2, we present the parameterizations on the VAT laminates with linearly varying fiber path angles. The mathematical expressions for the two common manufacturing constraints are also presented in this section. In Section 3, we briefly explain the buckling analysis of the stiffened plates subjected to

various in-plane end-shortening displacements using an approach that models the plate and the stiffeners independently. Section 4 summarizes the buckling load maximization problem. Optimization design variables, objective functions, and constraints are presented in this section. Different optimization results are compared and discussed in Section 5. The last section, Section 6, concludes the work.

## 2. Variable Angle Tow Laminates

A linearly varying fiber path angle [33] is considered for the varying angle fiber path reference line of each tape as it has been studied by many researchers for manufacturable VAT laminates [2,21,22]. The reference path is considered as the centerline of the tape. The fiber path angle varies linearly along the $x$-axis as shown in Figure 1a, whose expression is given as:

$$\theta = \begin{cases} \theta_0 + \dfrac{\theta_1 - \theta_0}{d}x, & 0 \leq x \leq d \\ \theta_0 - \dfrac{\theta_1 - \theta_0}{d}x, & -d \leq x \leq 0 \end{cases} \tag{1}$$

where $\theta_0$ and $\theta_1$ are, respectively, fiber path angles at $x = 0$ and $x = d$, and $d$ is a distance along the plate's length direction, i.e., $x$-axis. A rigid axis rotation of the whole lamina is also considered to enhance the design space of the laminate configurations. When there is a rigid rotation of the original coordinate system about the origin by an angle of $\phi$, the fiber path angle is expressed as:

$$\theta = \begin{cases} \phi + \theta_0 + \dfrac{\theta_1 - \theta_0}{d}x', & 0 \leq x' \leq d \\ \phi + \theta_0 - \dfrac{\theta_1 - \theta_0}{d}x', & -d \leq x' \leq 0 \end{cases} \tag{2}$$

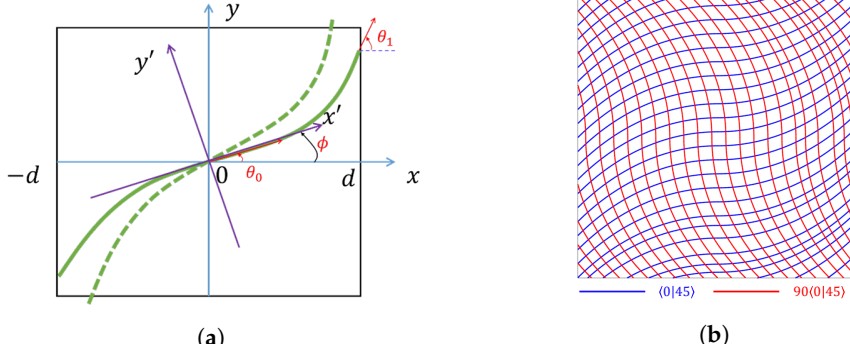

|     |     |
| :-: | :-: |
| (a) | (b) |

**Figure 1.** Linearly varying fiber path angle with a rigid rotation for a VAT lamina [20]. (**a**) Linearly varying fiber path angle; (**b**) two representative linearly varying VAT laminates.

For a single layer, the linearly varying fiber path angle along the $x$-axis as shown in Figure 1a is denoted as $\Theta = \langle \theta_0 | \theta_1 \rangle$. When there is a rigid rotation angle of the reference coordinate system, $\phi$ (see Figure 1a), the fiber path angle is denoted as $\Theta = \phi \langle \theta_0 | \theta_1 \rangle$. For $2n$-layer symmetric and balanced VAT laminates, the laminate configurations are denoted as $[(\pm \langle \theta_0 | \theta_1 \rangle)_n]_s$. When there is a rigid rotation of the whole laminate, the laminate configuration is denoted as $[(\phi \pm \langle \theta_0 | \theta_1 \rangle)_n]_s$. Figure 1b shows two examples of the linearly varying fiber path angles laminate configurations. The fiber path (in red) is obtained by rotating the fiber path (in blue) about the plate's center by 90 degrees. In this study, the new reference coordinate system, $x'y'$, is obtained by rotating the original reference coordinate system, $xy$, about the plate's center. The lines shown in Figure 1b are reference path lines for the tapes, i.e., the median line of each tape.

### 2.1. Manufacturing Constraint on Minimal AFP Head Turning Radius

The AFP head turning radius requires a curvature constraint for the fiber path; otherwise, there is a wrinkling or an upfolded tow generated in the placed tape [16]. A typical fiber placement machine generally lays courses with 4–32 tow each, which can be individually cut and reinitiated during the placement process. The tow width is normally 3.175 mm or 6.35 mm. The turning radius for the centerline for each course, i.e., reference line of each tape, leads to a constraint on the fiber path curve curvature (see Figure 2). The commonly used value of the turning radius is, $r_{min}$=0.635 m (25 inches) [23]. Therefore, the curvature of the fiber path curve is constrained as:

$$-\frac{1}{r_{min}} \leq \kappa \leq \frac{1}{r_{min}} \tag{3}$$

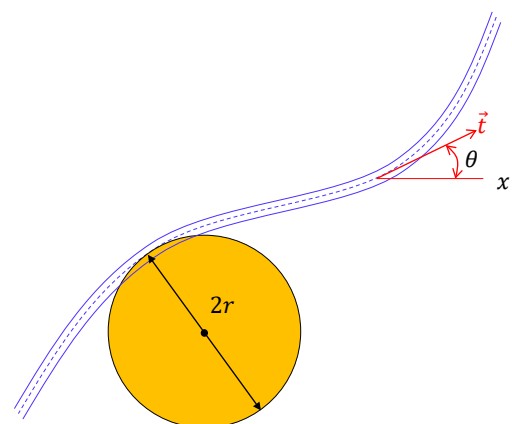

**Figure 2.** Curvature constraint for a towed tape reference line [20].

For a general expression, the fiber path orientation is a function of both $x$ and $y$ as $\theta(x, y)$. The unit tangential vector, $\vec{t}(\theta)$ (see Figure 2), can be expressed as:

$$\vec{t}(\theta) = \cos(\theta)\vec{i} + \sin(\theta)\vec{j} \tag{4}$$

The curvature is computed as the following [18]:

$$\kappa(x, y) = (\nabla \times \vec{t}(\theta)) \cdot \vec{k} = (\nabla \times (\cos(\theta)\vec{i} + \sin(\theta)\vec{j})) \cdot \vec{k}$$
$$= \frac{\partial \theta}{\partial x} \cos(\theta) + \frac{\partial \theta}{\partial y} \sin(\theta) \tag{5}$$

### 2.2. Parallel Fiber Tapes

Even if the curvature constraint is satisfied, during the tape placement, if the two adjacent tapes are not strictly parallel to each other, it is still possible to generate a gap (void), and/or an overlap in some portions (see Figure 3). The determination of two tapes parallels to each other or not at the two adjacent tapes' interface is computed by the two angles computed at the point in the tapes' interface.

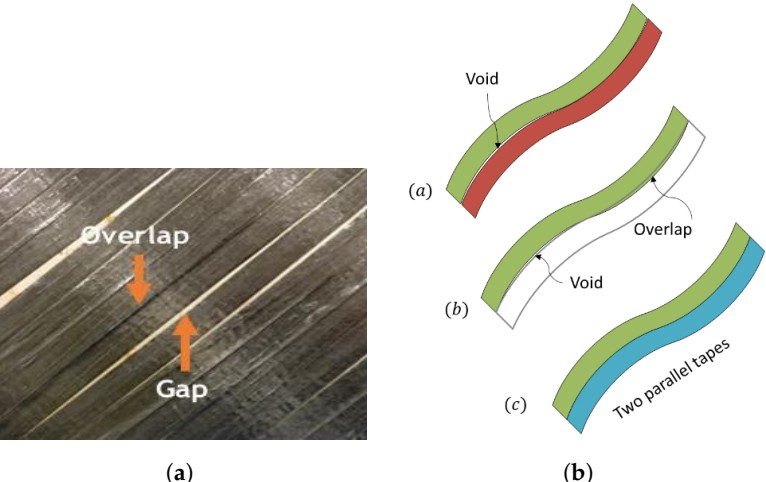

**(a)**　　　　　　　　　　**(b)**

**Figure 3.** Tapes gap/overlap generated in VAT laminates during AFP manufacturing. (**a**) gap/overlap representation [34]; (**b**) description of gap/overlap.

Following six steps are used for determining the parallelism between two adjacent tapes at any point:

(1) Select any point in the middle line of one tape (for example, Tape 1), $(x_0, y_0)$ (see Figure 4), compute the fiber path angle, $\theta_0$;

(2) Rotate the tangential vector computed using $\theta_0$, by 90°, to obtain the in-plane unit normal vector, $\vec{u}$;

(3) Compute the adjacent point node coordinate $(x, y)$ based on present point coordinate $(x_0, y_0)$, $\vec{u}$, and the tape width, $d_f$: $(x, y) = (x_0, y_0) + \vec{u}\dfrac{d_f}{2}$;

(4) Compute the fiber path angle, $\theta$, for the point $(x, y)$ (obtained in Step (3)) using the prescribed fiber path angle expression. Repeat Step (2) by rotating the new tangential vector at the point of $(x, y)$ by 90° for a new unit normal vector, $\vec{v}$;

(5) Compute the angle between the two normal vectors: $\sin(\Delta\alpha) = \dfrac{\vec{u} \times \vec{v}}{|\vec{u}||\vec{v}|} \cdot \vec{k}.$

(6) Repeat Steps (2)–(5) for another angle $\Delta\alpha$ through rotating the tangential vector by $-90°$ in Step (2).

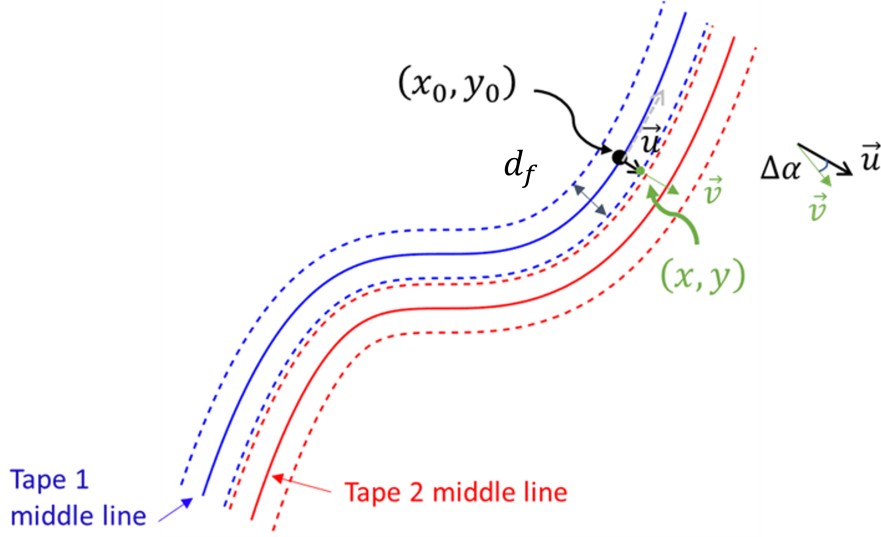

**Figure 4.** Parameterization of parallelism of two adjacent tapes.

If the two tapes are parallel to each other, $\Delta\alpha = 0$. In the present work, we constrain the angle difference to a small value of $1°$:

$$|\Delta\alpha| \leq 1° \tag{6}$$

In this study, the chosen tape width is $d_f = 3.175$ mm [16]. The minimum element size for the plate with a cutout is around 3 mm. This means that one tape may cover the whole element that has the minimal element size. For tape parallilism determination, we employed another model obtained by discretizing the whole VAT laminated panel into a high resolution mesh where the size for all elements is the same. Each element size is same or close to the tape width. The tape parallelism is checked for each element to ensure $|\Delta\alpha| \leq 1°$. Equation (6) is computed for all elements to determine the parallelism between the two adjacent tapes for the high-resolution mesh.

### 2.3. Material Properties

The material properties for the composite plate and the stiffeners are same, which are shown in Table 1.

**Table 1.** Material Properties [29,35].

| $E_1$ (GPa) | $E_2$ (GPa) | $\nu_{12}$ | $G_{12}$ (GPa) | $G_{13} = G_{23}$ (GPa) | Layer Thickness (mm) |
|---|---|---|---|---|---|
| 181.00 | 10.27 | 0.28 | 7.17 | 4.00 | 0.1272 |

## 3. Buckling Analysis Governing Equations

Consider a stiffened composite plate with a central cutout as shown in Figure 5. There are two straight blade stiffeners placed along the $x$-axis. The plate thickness is $t_p$. The blade stiffener cross section width and height are, respectively, denoted as $b_s$ and $h_s$. In this study, $b_s = t_p$, and the stiffener depth ratio is assumed to be $h_s/b_s = 5$. A large depth ratio for the stiffener might cause stiffener lateral buckling [36,37]. Concentric stiffeners are considered in the present work. This means that the stiffener centroid is on the plate median plane.

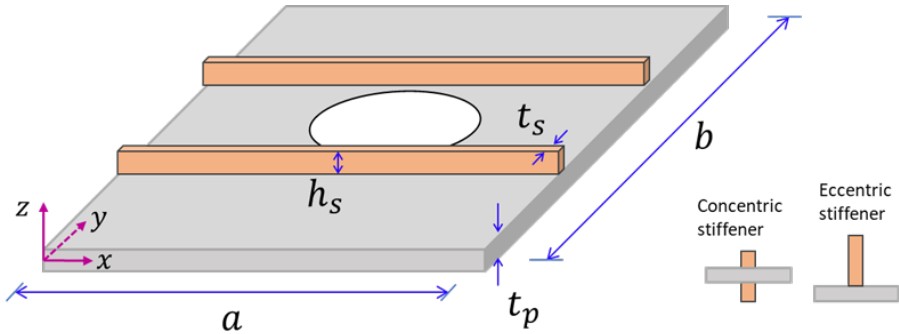

**Figure 5.** Stiffened plate with a central cutout.

For a general boundary condition, the finite element method is employed in this work to study the buckling and stress responses of the stiffened VAT laminated plates under various in-plane boundary conditions. Stress concentration is commonly seen in the regions on the plate near the cutout. It is necessary to use a fine mesh near the cutout for the plate's finite element model to capture the accurate stress/strain distribution. As explained previously, when considering the stiffener layout design, the changes in the stiffener shape and placement result in a change in the whole finite element mesh of the stiffened VAT laminated plate with cutouts. In addition, traditional finite element analysis requires that the mesh nodes at the interface between the plate and the stiffeners should coincide (see Figure 6c). All of these will lead to some difficulties in determining the converged mesh for the stiffened plate with cutouts for its subsequent structural analysis

in the automatic optimization study. To avoid these inconveniences, the plate and the stiffeners were modeled separately as seen in Figure 6b. The displacement compatibility condition is considered at the interface between the plate and the stiffener for the final model of the stiffened plate with a cutout. This independent modeling of the plate and the stiffeners allows us to use converged mesh for the plate (see Figure 6a) in both fiber path orientation optimization and stiffener shape optimization. This approach has been studied in our previous works [29,32], which will not be shown here in details.

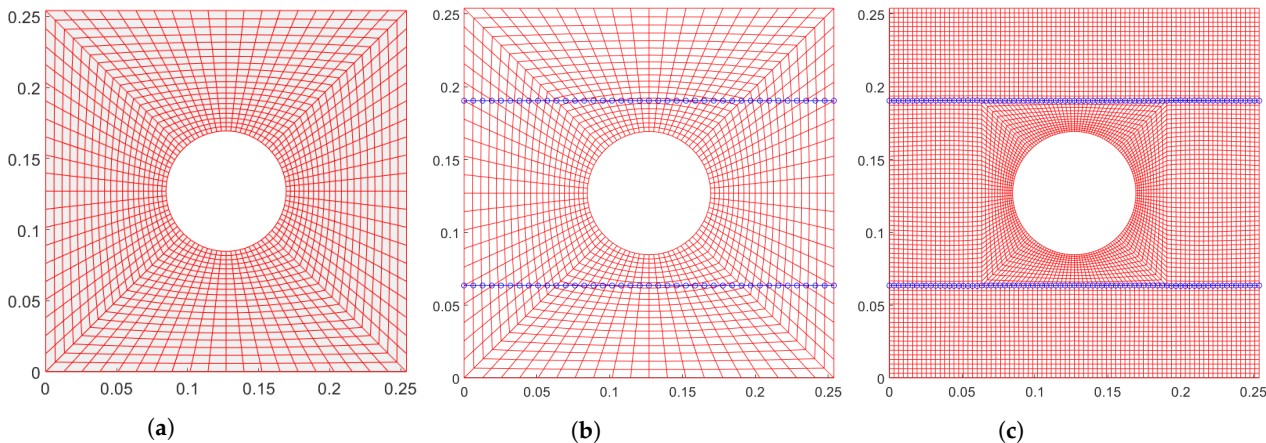

(a)            (b)            (c)

**Figure 6.** Finite element models of a stiffened VAT laminated plate with a central cutout; stiffener mesh (in blue). (**a**) Mesh of the plate with a cutout; (**b**) present finite element model; (**c**) NASTRAN finite element model.

### 3.1. Prebuckling Static Analysis

The buckling responses of the stiffened VAT laminated plate with cutouts under an in-plane end-shortening are studied. There are two reasons that we use a displacement control for studying the buckling responses of the stiffened VAT laminated plates: the displacement at the panel's edges can be easily obtained from the global aircraft design based on previously studied aircraft global/local design analysis optimization framework [38], and the displacement control is also easier to achieve in the laboratory. There are two steps considered in studying the buckling responses of the stiffened plate with a central cutout. The first step is to conduct a prebuckling static analysis to determine the in-plane stress resultants under the given initial displacements. The second step is to perform a bucking eigenvalue computation analysis. The prebuckling static analysis equation of the stiffened VAT laminated plates under initial displacements is:

$$\left[K_p + K_s\right]\{d_p\} = \{F\} \tag{7}$$

where $d_p$ is the plate's nodal displacement vector, and $F$ is the applied load force vector; $K_p$ and $K_s$ are the elastic stiffness matrices for the plate and the stiffeners, respectively. Their element stiffness matrices are expressed in the following:

$$
\begin{aligned}
K_p^e &= \int_{-1}^{+1}\int_{-1}^{+1} B_p^T D_p^T B_p \det J_p \mathrm{d}\xi \mathrm{d}\eta \\
K_s^e &= \int_{-1}^{+1} N_{sp}^T T_s^T B_s^T D_s^T B_s T_s N_{sp} \det J_s \mathrm{d}\xi
\end{aligned}
\tag{8}
$$

where $N_{sp}$ is the stiffener nodal displacement interpolation matrix, which can be found in previous work [32]; $J_p$ and $J_s$ are the Jacobians for the panel and the stiffener, respectively. Eight-noded plate element and three-noded beam element for the plate and the stiffener are, respectively, use in the finite element modeling.

The displacement field for the plate is summarized as:

$$d_p = \left\{ \begin{array}{c} \Delta d \\ d \end{array} \right\} \tag{9}$$

where the in-plane end-shortening vector is $\Delta d$. Equation (7) can be rewritten as:

$$[K]\{d_p\} = \left[ \begin{array}{cc} (K_{11})_{k \times k} & (K_{12})_{k \times l} \\ (K_{21})_{l \times k} & (K_{22})_{l \times l} \end{array} \right] \left\{ \begin{array}{c} \Delta d \\ d \end{array} \right\} = \left\{ \begin{array}{c} f \\ 0 \end{array} \right\} \tag{10}$$

where $K = K_p + K_s$, $k$ is the number of degrees of freedom with known displacements, $l$ is the number for the degrees of freedom with unknown displacements, and $f$ is the nodal force vector applied in the plate's edge. The expression for the unknown displacement, $d$ in Equation (10), is

$$d = -K_{22}^{-1}K_{21}\Delta d \tag{11}$$

Stress recovery for both the plate and the stiffeners can be obtained as:

$$\sigma_p = Q_p \varepsilon_p = Q_p B_p d_p \tag{12a}$$

$$\sigma_t = Q_s \varepsilon_t = Q_s B_s T_s N_{sp} d_p \tag{12b}$$

where $Q_p$ and $\varepsilon_p$ have different values when they are computed at different Gaussian points for each element. In this work, the averaged stress for each element in each layer of the laminates is used to compute the geometric stiffness matrices for the plate and the stiffeners.

### 3.2. Buckling Load Factor

The buckling load factor, $\lambda_b$, can be obtained through an eigenvalue computation as:

$$\left[ (K_p + K_s) + \lambda_{cr}(K_{Gp} + K_{Gs}) \right]\{\Psi_p\} = 0 \tag{13}$$

In the above equations, $\lambda_{cr}$ and $\Psi_p$ are buckling eigenvalue and buckling mode shape, respectively; $K_{Gp}$ and $K_{Gs}$ are the geometric stiffness matrices for the plate and the stiffeners, respectively. The element geometric stiffness matrices, $K_{Gp}^e$ and $K_{Gs}^e$, respectively, for assembling these global matrices are:

$$\begin{aligned} K_{Gp}^e &= \int_{-1}^{+1} \int_{-1}^{+1} \left( B_p^{NL} \right)^T N_p B_p^{NL} \det J_p \mathrm{d}\xi \mathrm{d}\eta \\ K_{Gs}^e &= \int_{-1}^{+1} N_{sp}^T T_s^{T} \left( B_s^{NL} \right)^T N_s B_s^{NL} T_s N_{sp} \det J_s \mathrm{d}\xi \end{aligned} \tag{14}$$

where $N_p$ and $N_s$ are stress resultants matrices for the plate and the stiffener, respectively; the expressions for the matrices given in Equation (14) and numerical approaches used to obtain these elemental stiffness matrices can be found in [32,39].

## 4. Optimization Problem Statement

The chosen stiffened plate size is $a = b = 0.254$ m, whose panel size is commonly used [29,40]. A central circular cutout is considered whose radius is $r = a/6$. The scope of this work is to study the effect of varying the angle tow fiber path on the structural performance of a stiffened plate with a central cutout. Therefore, the cutout size and its location are fixed in the following studies under different boundary conditions. A total of 16-layer symmetric, balanced laminates are considered for the VAT laminated plate, $[(\pm\Theta)_4]_s$. The thick laminates lead to a small in-plane deformation under the studied, applied in-plane displacement, which does not affect the geometry shape significantly in the following buckling analysis. This means that the undeformed model is considered in the buckling analysis. The buckling load maximization studies for the stiffened VAT laminates

without considering the AFP manufacturing constraints on the fiber path angles will be first studied. Those results will be used as the base results in the following comparison studies to investigate the effect of the AFP manufacturing constraints on the bucking responses. Different laminate configurations are listed in Table 2. Quasi-isotropic laminates, also named as "black metal" composite, are the most common type of composite structures as used in the practical applications [41], which is considered as a base model in the present study.

**Table 2.** Summary of design optimization cases.

| Fibre Path | Description |
|---|---|
| QI | Quasi-isotropic laminates, $[(\pm 45/0/90)_2]_s$ * |
| SF | Straight fiber path laminates |
| LV-X | Linearly varying fiber path along the $x$-axis |
| LV-XY | Linearly varying fiber path along the $x$-axis with a rigid rotation |

* The superscript degree sign for the fiber path angle in the laminate configuration is ignored for convenience.

### 4.1. Optimization Design Variables

The design variables used in the optimization are the fiber path angles (see $\Theta$ defined in Sections 2). In addition, the stiffeners normalized location is considered to tailor the buckling mode shape for an increased buckling load. For simplicity, symmetric stiffeners are considered for the the present model. The normalized stiffener location, $\bar{x}$, is shown in Figure 7. The two parallel stiffeners are considered for the stiffened plate under axial and bending displacement boundary condition cases and the four orthogonal stiffeners are considered for the stiffened plate under in-plane pure shear displacement boundary condition case. All the design variables are summarized in Table 3. For the quasi-isotropic laminate configuration (see Table 2), the design variable is only the stiffener normalized location.

**Table 3.** Summary of Design Variables.

| Design Variable | Lower Bound | Upper Bound |
|---|---|---|
| Fiber path angle configuration, $\Theta$, (degrees) | $-90$ | 90 |
| Stiffener normalized location, $\bar{x}$ | 0 | 1/3 |

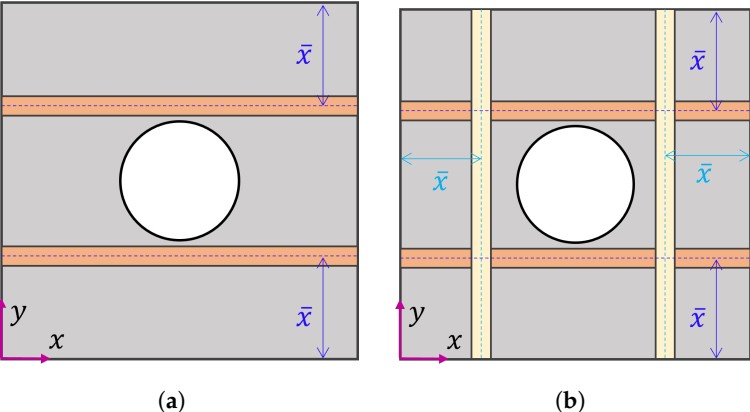

      (**a**)             (**b**)

**Figure 7.** Stiffened plate with different number of stiffeners. (**a**) two parallel stiffeners; (**b**) four orthogonal stiffeners.

### 4.2. Optimization Constraints

The design constraints considered in the optimization are two manufacturing constraints as explained in Sections 2.1 and 2.2. Based on Equations (3) and (6), all the constraints are normalized and are expressed as:

$$
\begin{aligned}
g_1 &= \kappa / \left( \frac{1}{r_{min}} \right) - 1 \le 0 \\
g_2 &= -\kappa / \left( \frac{1}{r_{min}} \right) - 1 \le 0 \\
g_3 &= -\Delta\alpha - 1 \le 0 \\
g_4 &= \Delta\alpha - 1 \le 0
\end{aligned}
\tag{15}
$$

### 4.3. Optimization Objective Function

The optimization is to maximize the buckling load for the stiffened VAT laminates subjected to in-plane end-shortenings. For the stiffened plate subjected to an in-plane axial shortening, the critical buckling load, $P_{cr}$, is computed as:

$$
P_{cr} = \lambda_{cr} \left[ \int_0^b N_{xx}(a,y)\mathrm{d}y + \sum_{i=1}^n A_{s,i}\sigma_{s,i} \right]
\tag{16}
$$

where $\lambda_{cr}$ is the buckling load factor obtained from an eigenvalue computation using Equation (13); $N_{xx}(a,y)$ is the stress resultant at the panel edge, $x = a$, along the plate's width direction ($y$-axis); $A_s$ and $\sigma_{s,i}$ are the $i$th stiffener cross section area and the axial stress, respectively.

For the stiffened plate subjected to an in-plane shear displacement, the critical buckling load, $P_{cr}$, is computed as:

$$
P_{cr} = \lambda_{cr} \left[ \int_0^b N_{xy}(a,y)\mathrm{d}y + \sum_{i=1}^n A_{s,i}\tau_{s,i} \right]
\tag{17}
$$

where $N_{xy}(a,y)$ is the shear stress resultant at the panel edge, $x = a$, along the plate's width direction ($y$-axis); $\tau_{s,i}$ is the $i$th stiffener in-plane shear stress. For the stiffened plate under the in-plane axial and in-plane shear displacement load cases, the normalized buckling load parameter is used and it is expressed as:

$$
K_{cr} = \frac{P_{cr}a^2}{E_1 t_p^3 b}
\tag{18}
$$

For the stiffened plate subjected to an in-plane pure bending displacement, the critical buckling load, $M_{cr}$, is computed about the central line of the plate, $y = b/2$.

$$
M_{cr} = \lambda_{cr} \left[ \int_0^b N_{xx}(a,y)(y - \frac{b}{2})\mathrm{d}y + \sum_{i=1}^n A_{s,i}\sigma_{s,i}(y_{s,i} - \frac{b}{2}) \right]
\tag{19}
$$

where $y_{s,i}$ is the location of the $i$th stiffener. For the in-plane bending displacement load case, the normalized buckling load parameter is:

$$
K_{cr} = \frac{M_{cr}a}{E_1 t_p^3 b}
\tag{20}
$$

The optimization is conducted to maximize the normalized buckling parameter for each boundary condition. Note that the present optimization fitness function is nonconvex in terms of the fiber path orientations. As a result, the previously developed particle swarm optimization is employed [20,38]. The buckling load maximization of the stiffened VAT

laminated plate is converted to a minimization optimization problem of $(-K_{cr})$, which is expressed as:

$$
\begin{aligned}
\textbf{minimize} \quad & -K_{cr} \\
\textbf{w.r.t.} \quad & \Theta \text{ and } \bar{x} \\
\textbf{s.t.} \quad & -90° \le \Theta \le 90° \\
& 0 \le \bar{x} \le 1/3 \\
& g_i \le 0
\end{aligned}
$$

For a constrained particle swarm optimization (PSO) problem, the mostly common approach is to aggregate all constraints with the original cost function as an unconstrained optimization problem using a penalty approach. In the present study, only the manufacturing constraints are considered in the optimization problem. The objective function for the unconstrained PSO problem is:

$$
f = -K_{cr} + \sum_{i=1}^{ng} \lambda_i G_i \tag{21}
$$

where $ng$ is the number of violated constraints and $\lambda_i$ is the penalty factor for the violated constraint, $G_i$. When adding the violated constraints to the cost function, $G_i$ is written as:

$$
G_i = \begin{cases} g_i, & g_i > 0 \\ 0, & g_i \le 0 \end{cases} \tag{22}
$$

where $g_i$ is the $i$th normalized constraint, as given in Equation (15), a large value of the penalty factor is chosen for the normalized violated constraint, $\lambda_i = 1.0E+06$ [20].

## 5. Optimization Results

### 5.1. Load Case A: In-Plane Axial Displacement

The boundary condition for the stiffened plate with a central hole in the prebuckling static analysis is shown in Figure 8. During the prebuckling static analysis,

- $x = 0, u = \Delta$ and $x = a, u = -\Delta x$, where $\Delta x = 0.01$ mm
- $y = 0$ and $y = b, v = 0$;

In the buckling analysis, four edges are simply supported: $u = v = w = 0$.

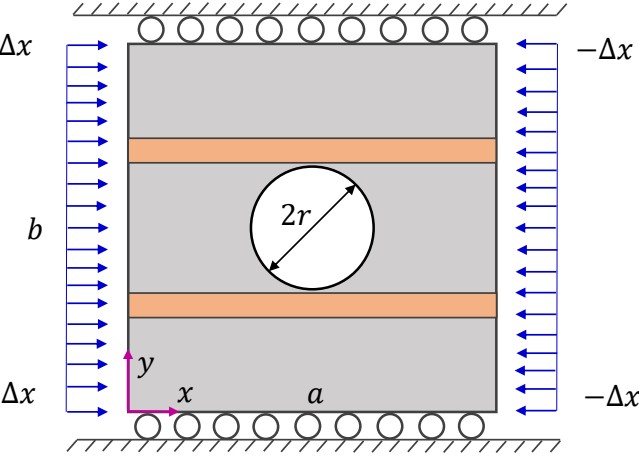

**Figure 8.** Composite stiffened plate with a central hole under uniform in-plane axial displacements.

### 5.1.1. Verification Studies

The present program has been verified for buckling analysis of stiffened VAT laminates without cutouts under in-plane axial loads [20]. To gain confidence in the present program

in computing the buckling load of the stiffened VAT laminated plate with a central cutout under various in-plane boundary conditions, further verification studies are considered. In the present verification study, two stiffeners are placed at $y = b/4$ and $y = 3b/4$ where $b$ is the width of the plate. An eight-layer symmetric laminate with hybrid fiber path orientations is considered for the plate in the verification study, $[(45)_2/45\langle\pm90|0\rangle]_s$. The stress resultants for the stiffened plate under uniform axial displacements are computed and compared against NASTRAN results. The stress resultant distributions for the three in-plane stress components are in a good agreement with NASTRAN results as seen in Figure 9. The present stiffener dominant axial stress distribution is also close to NASTRAN results, which are shown in Figure A1a in Appendix A for brevity. The present buckling mode results also match well with NASTRAN results as seen in Figure 10. All NASTRAN results reported in this paper are postprocessed using MATLAB (see NASTRAN models and the MATLAB subroutines in the Data Availability Statement section).

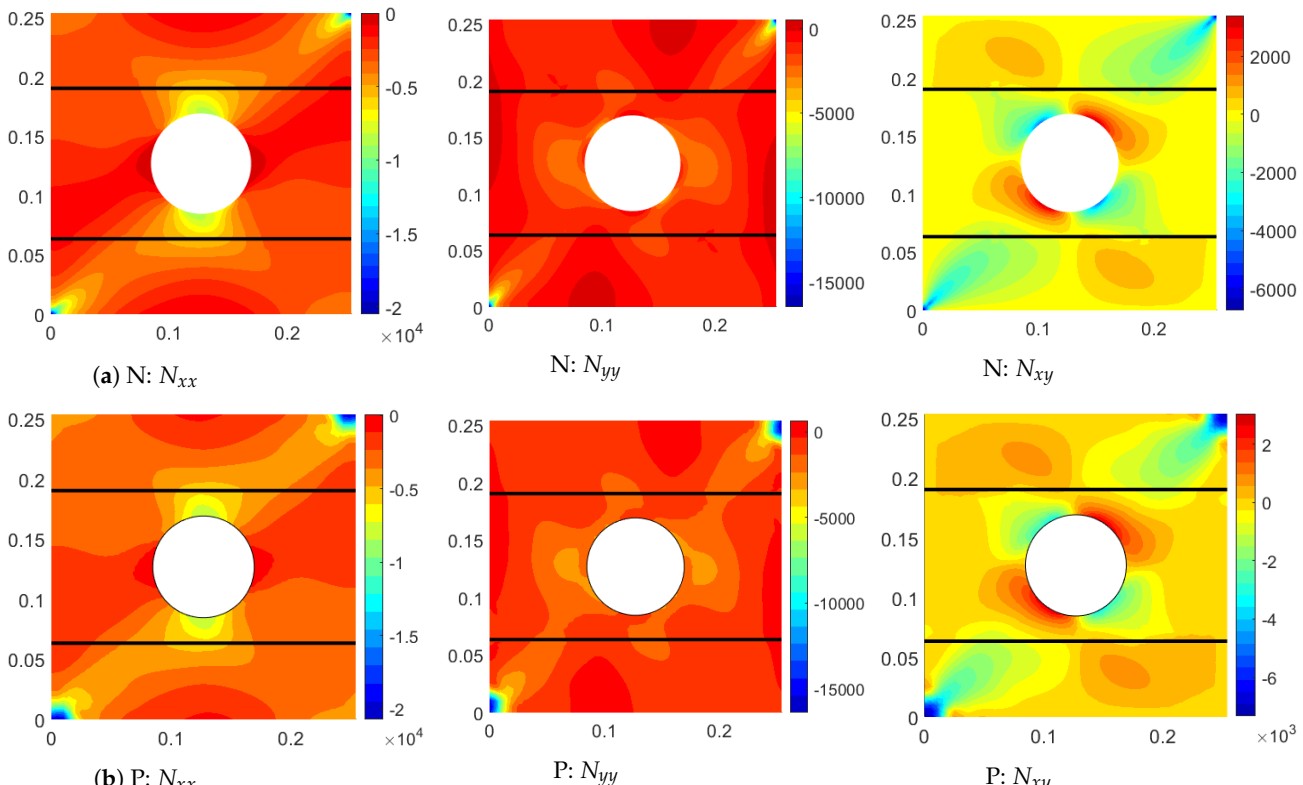

**Figure 9.** Comparison of stress resultants (unit: N/m) of the stiffened VAT laminates with a central hole under uniform axial displacements. (**a**) N: NASTRAN results; (**b**) P: Present results.

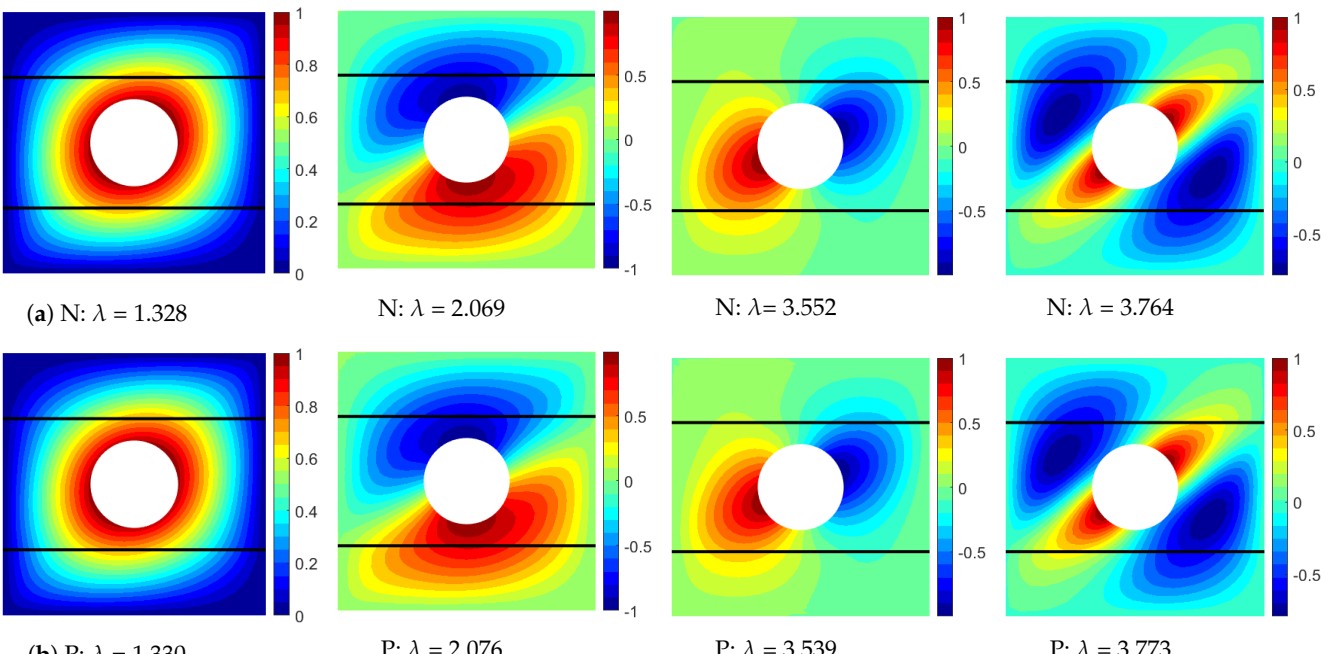

**(a)** N: λ = 1.328  N: λ = 2.069  N: λ= 3.552  N: λ = 3.764

**(b)** P: λ = 1.330  P: λ = 2.076  P: λ = 3.539  P: λ = 3.773

**Figure 10.** Comparison of buckling eigenvalue (λ) and buckling mode shapes of the stiffened VAT laminates with a central hole under uniform axial displacements. (**a**) N: NASTRAN results; (**b**) P: Present results.

5.1.2. Optimization without Manufacturing Constraints

Buckling load maximization is first conducted for the stiffened VAT laminated plate subjected to uniform axial displacements without considering AFP manufacturing constraints. The optimal configurations for different laminate configuration cases including the fiber path angles and stiffener normalized locations, and the buckling analysis results for the stiffened VAT laminates are shown in Table 4. The results show that the buckling load can be increased by 21.28% and 12.46%, respectively, when using linearly varying fiber path angle laminates with a 90-degree rotation for the plate comparing to the quasi-isotropic laminates and the optimal straight fiber path laminates results.

The optimal fiber paths for different laminates cases and the in-plane stress resultants and buckling mode results for these optimal models are summarized in Figure 11. It is seen that the stress concentration occurs near the hole's edge when using straight fiber path laminates. It is clearly observed that linearly varying fiber path angle laminates with a rigid rotation (LV-XY) can move the location for the maximum value of the dominant in-plane stress resultant (compression stress), $N_{xx}$, from the hole's edge to the panel support edges ($y = 0, b$) (see $N_{xx}$ in Figure 11d) for an increased buckling load. This phenomena is also observed in the study by Lopes et al. [10] where the VAT laminates tailor the stress distribution to improve the structural and buckling performance of the plate with cutouts..

**Table 4.** Optimal designs and results of the stiffened plate with a central cutout subjected to uniform axial displacement (w/o AFP manufacturing constraints).

| Case | Θ | $\bar{x}$ | $K_{cr}$ | Improvement [1] | Improvement [2] |
|------|---|-----------|----------|-----------------|-----------------|
| QI | $[(\pm45/0/90)_2]_s$ | 0.333 | 2.98 | – | – |
| SF | $[(\pm14.6)_4]_s$ | 0.311 | 3.21 | +7.84% | – |
| LV-X | $[(\pm\langle-6.3|33.0\rangle)_4]_s$ | 0.314 | 3.42 | +14.90% | 6.54% |
| LV-XY | $[(-90\pm\langle-42|90\rangle)_4]_s$ | 0.333 | 3.61 | +21.28% | 12.46% |

[1]: Compared to QI laminates results; [2]: compared to SF laminates results.

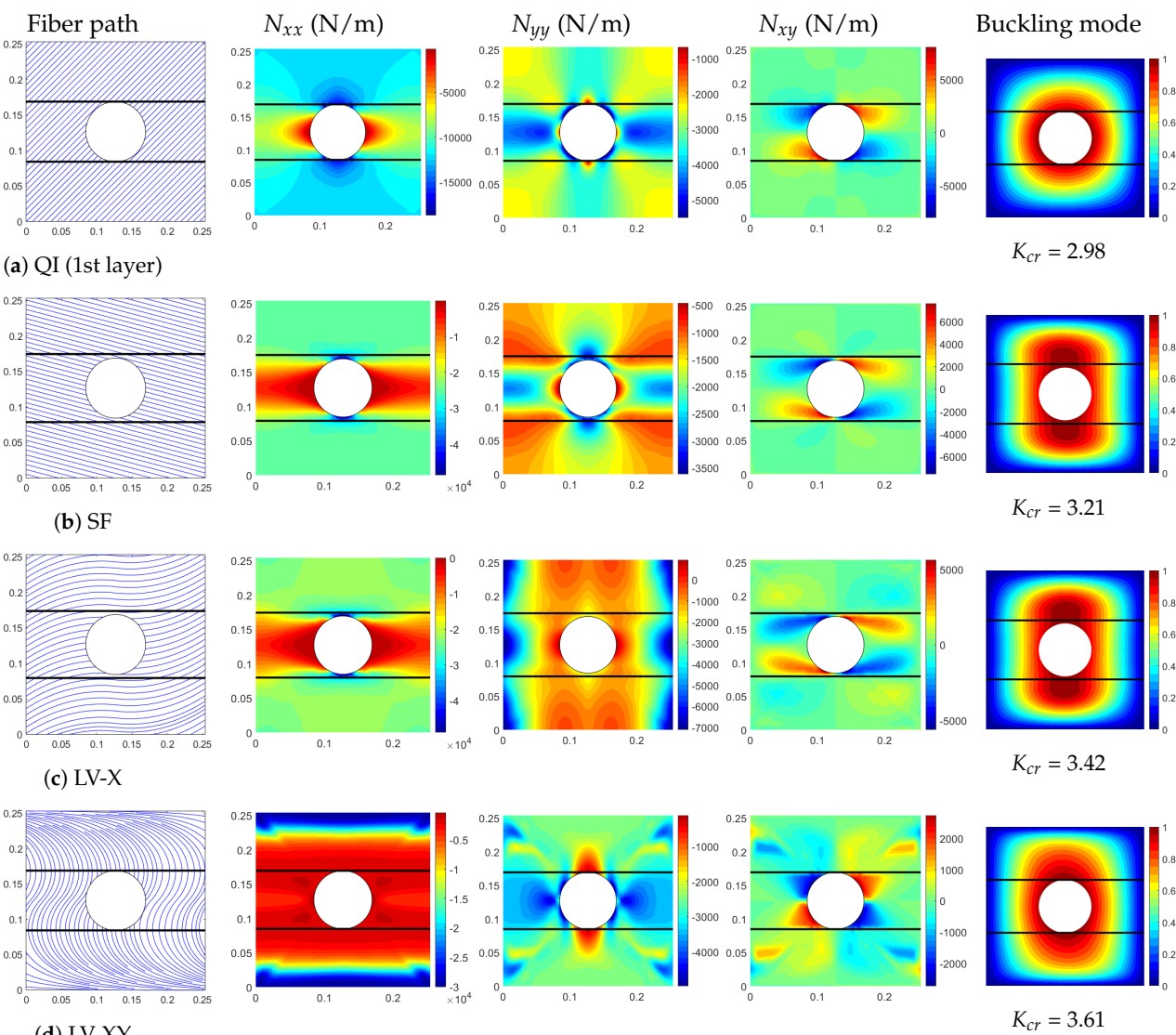

**Figure 11.** Comparison of optimal fiber paths, stress resultants and buckling modes for different study cases of the stiffened VAT laminates with a central hole under uniform axial displacements; the line drop in the fiber path plot does not mean the fiber cut in this study. (**a**) QI laminate design; (**b**) SF laminate design; (**c**) LV-X laminate design; (**d**) LV-XY laminate design.

### 5.1.3. Optimization with Manufacturing Constraints

The AFP manufacturing constraints are considered in the buckling load maximization studies. The optimal laminate configurations and the maximum buckling parameters for the four different laminates cases for the stiffened VAT laminates with a central hole under uniform axial displacements are shown in Table 5. The optimal fiber path, the in-plane stress resultants, and the buckling mode result for VAT laminates are shown in Figure 12. Since there are no AFP manufacturing constraints for the straight fiber path laminates, the two laminate configuration results for the cases of QI and SF are not shown in Figure 12.

**Table 5.** Optimal designs and results of the stiffened plate with a central cutout subjected to uniform axial displacements (w/ AFP manufacturing constraints).

| Case | $\Theta$ | $\bar{x}$ | $K_{cr}$ | Improvement [1] | Improvement [2] |
|---|---|---|---|---|---|
| QI | $[(\pm 45/0/90)_2]_s$ | 0.333 | 2.978 | – | – |
| SF | $[(\pm 14.6)_4]_s$ | 0.311 | 3.21 | +7.84% | – |
| LV-X | $[(\pm \langle 9.56\vert 21.19 \rangle)_4]_s$ | 0.313 | 3.307 | +11.05% | 3.02% |
| LV-XY | $[(-0.88 \pm \langle -9.49 \vert -21.11 \rangle)_4]_s$ | 0.333 | 3.309 | +11.11% | 3.08% |

[1]: Compared to QI laminates results; [2]: compared to SF laminates results.

Table 5 shows that the AFP manufacturing constraints have a significant influence on the optimal design results for all the VAT laminate configurations when comparing to the design results without considering AFP manufacturing constraints (see Table 4). The optimal linearly varying fiber path angle laminates (LV-XY case) only increase the buckling load by 11.11% compared to the quasi-isotropic laminate design, and a mere 3.08% increase in the buckling load comparing to the optimal straight fiber path laminate designs. In this design optimization study, the fiber path radius of curvature constraint is active for the two VAT laminate cases. This demonstrates that when considering AFP manufacturing constraints, there would be only a limited increase in the buckling load when using VAT laminates for the stiffened plate with a central hole under the in-plane axial displacement boundary condition. The optimal fiber path, in-plane stress resultants, and buckling mode shapes for the two optimal varying fiber path laminates are shown in Figure 12. The optimal fiber paths for these two laminate cases are similar when considering the AFP manufacturing constraints. The optimal stiffener locations, $\bar{x}$, for the two varying fiber path angle laminate cases are almost same as those obtained without considering AFP manufacturing constraints (see $\bar{x}$ in Tables 4 and 5). The AFP manufacturing constraints result in the buckling load 9–10% less than the one without considering AFP manufacturing constraints.

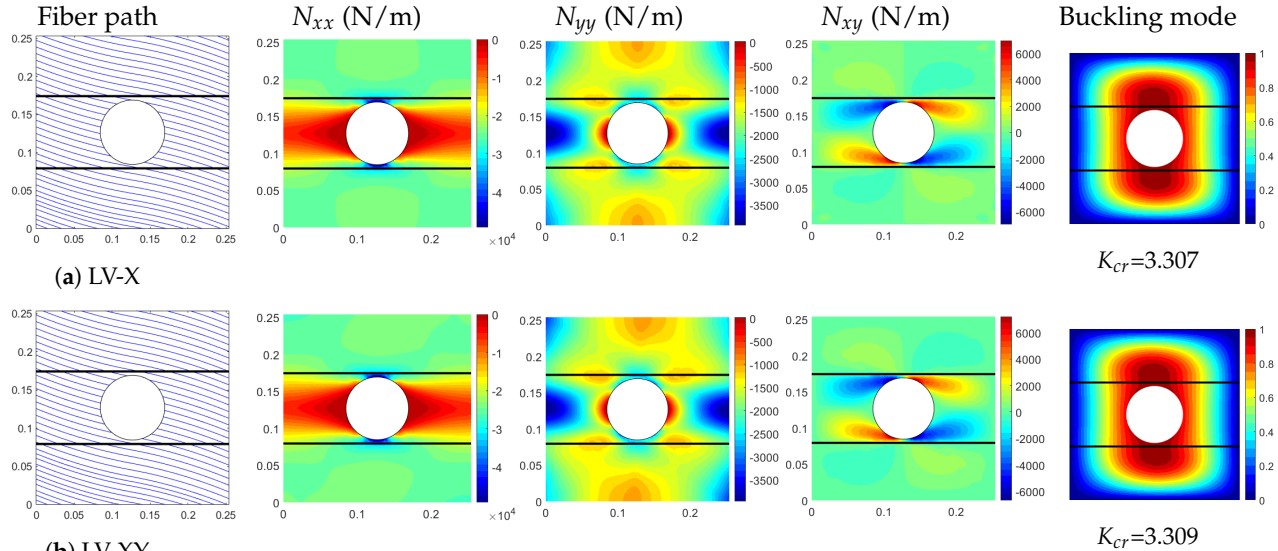

**Figure 12.** Comparison of optimal fiber paths, stress resultants, and buckling modes for different study cases of the stiffened VAT laminates with a central hole under uniform axial displacement considering AFP manufacturing constraints. (**a**) LV-X laminate design; (**b**) LV-XY laminate design.

*5.2. Load Case B: In-Plane Shear Displacement*

In this load case, we consider in-plane shear displacements as shown in Figure 13. The in-plane displacements at the four edges are expressed as:

- $x = 0$: $u = \dfrac{\Delta x}{(b/2)}(y - b/2)$ and $v = -\Delta y$;

- $x = a$: $u = \dfrac{\Delta x}{(b/2)}(y - b/2)$ and $v = \Delta y$;

- $y = b$: $u = \Delta x$ and $v = \dfrac{\Delta y}{(a/2)}(x - a/2)$;

- $y = 0$: $u = -\Delta x$ and $v = \dfrac{\Delta y}{(a/2)}(x - a/2)$.

where $\Delta x = \Delta y = 0.01$ mm is considered. During the buckling analysis, the plate's four edges are simply supported, $u = v = w = 0$.

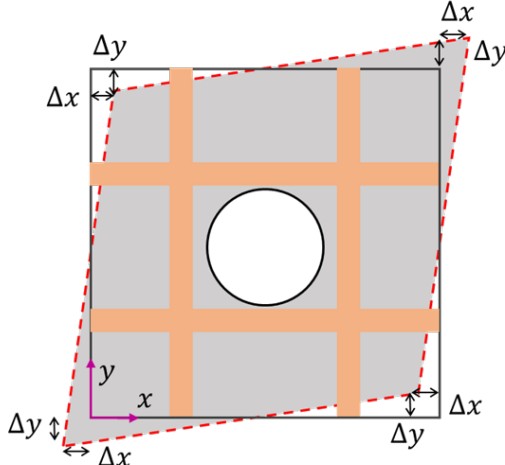

**Figure 13.** Composite stiffened plate under in-plane shear displacements.

### 5.2.1. Verification Studies

In this verification study, four orthogonal stiffeners are placed at $x = a/4$, $x = 3a/4$, $y = b/4$, and $y = 3b/4$ where $a$ and $b$ are length and the width of the plate, respectively. The plate's geometry and laminate configurations are same as the laminated model studied in Section 5.1.1. The buckling mode results for the stiffened VAT laminates under the in-plane shear displacements computed using the present program match well with NASTRAN results, as seen in Figure 14. The stress resultant distributions for the three in-plane stress components are also in a good agreement with NASTRAN results, as shown in Figure 15. The stiffener dominant axial stress distribution is also close to NASTRAN results (see Figure A1b in Appendix A).

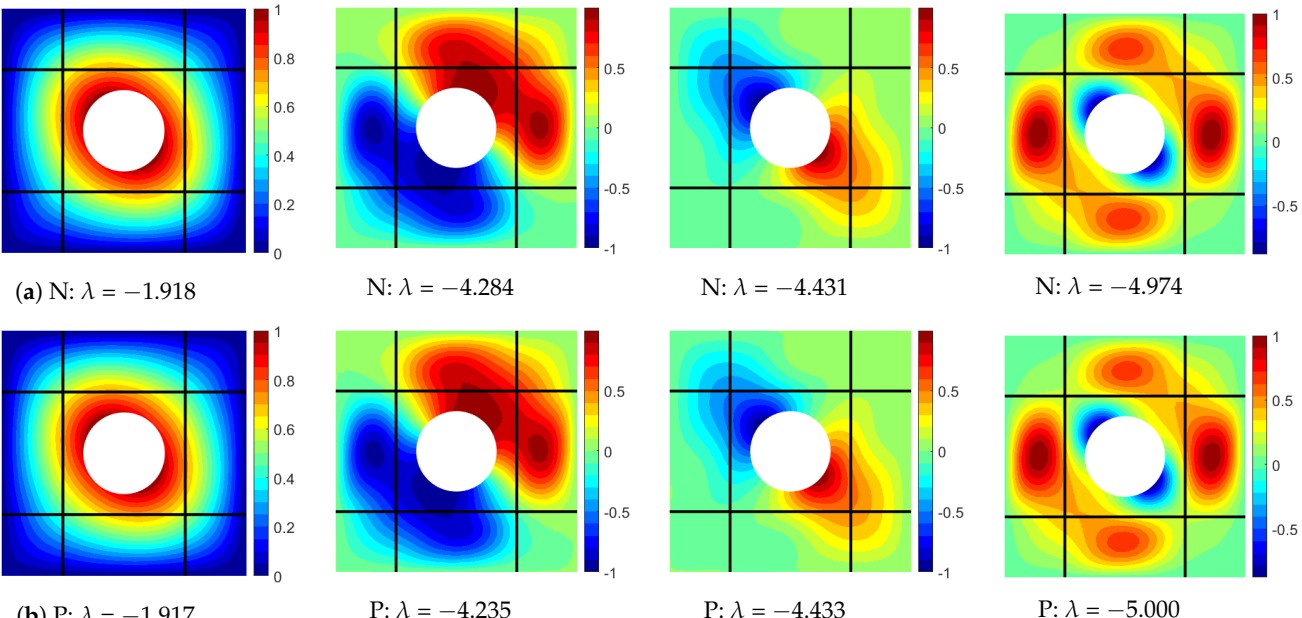

**Figure 14.** Comparison of buckling eigenvalue ($\lambda$) and buckling mode shapes of the stiffened VAT laminates with a central hole under in-plane shear displacements. (**a**) N: NASTRAN results; (**b**) P: Present results.

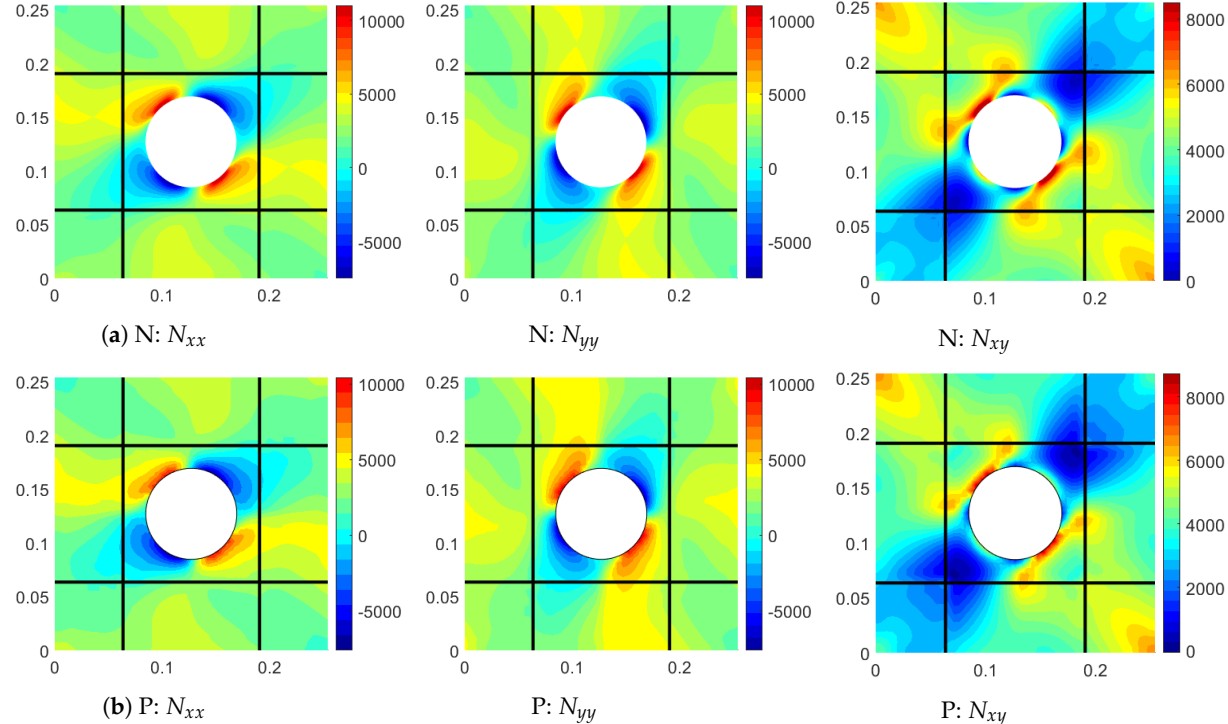

**Figure 15.** Comparison of stress resultants (unit: N/m) of the stiffened VAT laminates with a central hole under in-plane shear displacements. (**a**) N: NASTRAN results; (**b**) P: Present results.

### 5.2.2. Optimization without Manufacturing Constraints

Table 6 shows the buckling results for the stiffened VAT laminates with a central cutout under in-plane shear displacements without considering the AFP manufacturing constraints. It is clearly seen that using varying fiber path angle laminates (LV-XY case) can increase the buckling load significantly by up to 19.76% and 12.51%, respectively, as compared to quasi-isotropic laminates and optimal straight fiber path laminates.

**Table 6.** Optimal designs and results of the stiffened plate with a central cutout subjected to in-plane shear displacements (w/o AFP manufacturing constraints).

| Case | $\Theta$ | $\bar{x}$ | $K_{cr}$ | Improvement [1] | Improvement [2] |
|------|----------|-----------|----------|-----------------|-----------------|
| QI | $[(\pm 45/0/90)_2]_s$ | 0.308 | 9.16 | – | – |
| SF | $[(\pm 53.37)_4]_s$ | 0.307 | 9.75 | +6.44% | – |
| LV-X | $[(\pm \langle 90.0|50.73\rangle)_4]_s$ | 0.275 | 10.82 | +18.12% | +10.97% |
| LV-XY | $[(-90.0 \pm \langle -68.26|48.14\rangle)_4]_s$ | 0.318 | 10.97 | +19.76% | +12.51% |

[1]: Compared to QI laminates results; [2]: compared to SF laminates results.

Figure 16 shows the optimal fiber paths, in-plane stress resultant distribution, and the buckling mode shapes for the four designs. It is seen that the buckling mode shapes for the four different laminate configuration cases are almost identical where the buckling mode deformation peak occurs near the hole's edge. For the model under in-plane shear displacements, the location of the maximum dominant shear stress resultant, $N_{xy}$ moves away from the hole's edge to the panel vertices when using VAT laminates for the plate. This demonstrates again that it is the load redistribution capacity of the VAT laminates that increases the structural buckling load.

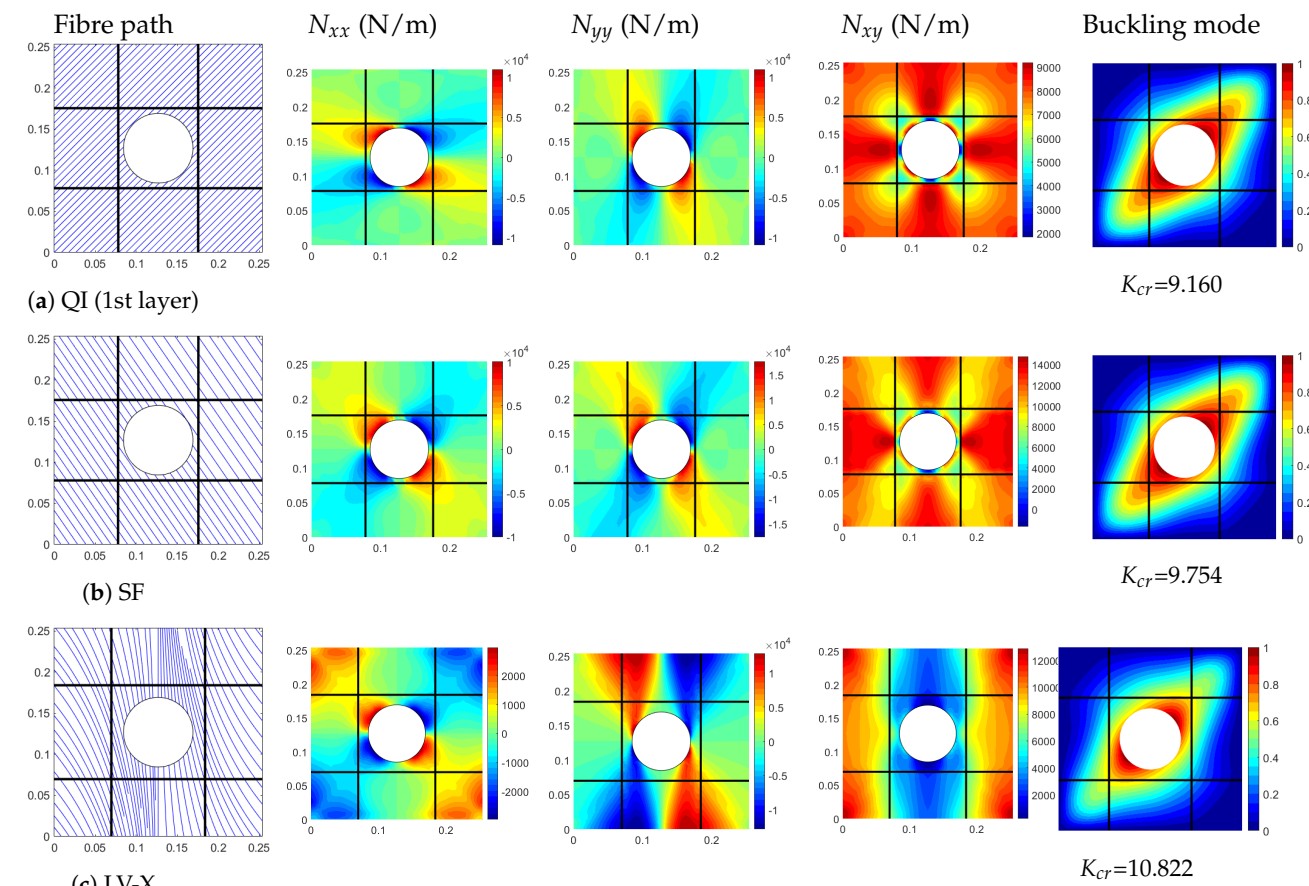

**Figure 16.** *Cont.*

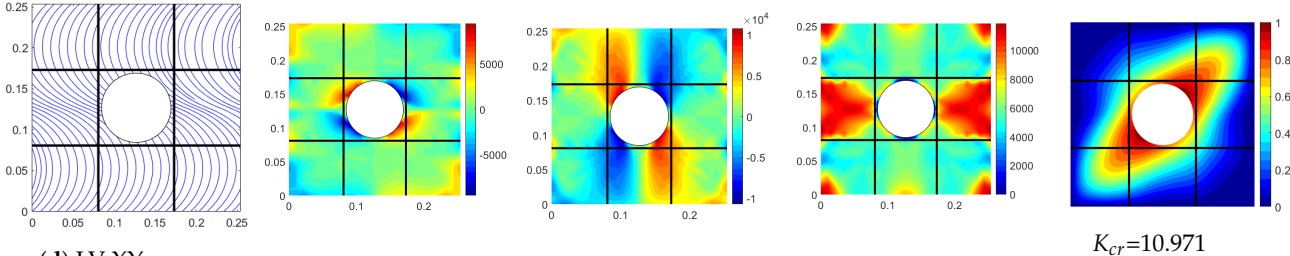

**(d)** LV-XY

$K_{cr}$=10.971

**Figure 16.** Comparison of optimal fibre paths, stress resultants and buckling modes for different study cases of stiffened plate under in-plane shear displacements. (**a**) QI laminate design; (**b**) SF laminate design; (**c**) LV-X laminate design; (**d**) LV-XY laminate design.

### 5.2.3. Optimization with Manufacturing Constraints

When considering the AFP manufacturing constraints on fabricating VAT laminates, Table 7 shows that using the varying fiber path angle laminates with a rigid rotation for the whole laminate (LV-XY) can increase the buckling load of the stiffened plate by 16.48% and 9.43%, respectively, as compared to the quasi-isotropic and optimal straight fiber path laminate results. There is around 3% reduction in the buckling load improvement when compared to the improvement for the design obtained without considering AFP manufacturing constraints, as seen in Table 6. This shows that the AFP manufacturing constraints have an impact on optimized VAT laminates design under the shear displacement boundary condition, although not as much as that for the stiffened plate in the presence of the in-plane axial load (see Table 5). The fiber path radius of curvature constraint is active in this optimization study.

**Table 7.** Optimal designs and results of the stiffened plate with a central cutout subjected to in-plane shear displacements (w/ AFP manufacturing constraints).

| Case | $\Theta$ | $\bar{x}$ | $K_{cr}$ | Improvement [1] | Improvement [2] |
|------|----------|-----------|----------|-----------------|-----------------|
| QI | $[(\pm 45/0/90)_2]_s$ | 0.308 | 9.16 | – | – |
| SF | $[(\pm 53.37)_4]_s$ | 0.307 | 9.75 | +6.44% | – |
| LV-X | $[(\pm \langle 90.0 \vert 64.0 \rangle)_4]_s$ | 0.283 | 10.49 | +14.52% | +7.59% |
| LV-XY | $[(0.43 \pm \langle -90.00 \vert -64.08 \rangle)_4]_s$ | 0.300 | 10.67 | +16.48% | +9.43% |

[1]: Compared to QI laminates results; [2]: Compared to SF laminates results.

The two linearly varying fibre path angle laminates, LV-X and LV-XY, have similar optimal laminate configurations in the present study. However, the optimal stiffener layouts, $\bar{x}$ for the laminate configuration design of LV-XY are slightly different. Figure 17 shows the optimal fiber paths for different VAT laminate configurations. The dominant stress resultant maximum value, $N_{xy}$, for the two linearly varying fiber path angle laminate configurations, LV-X and LV-XY, is shifted from the hole's edge to the plate's four vertices.

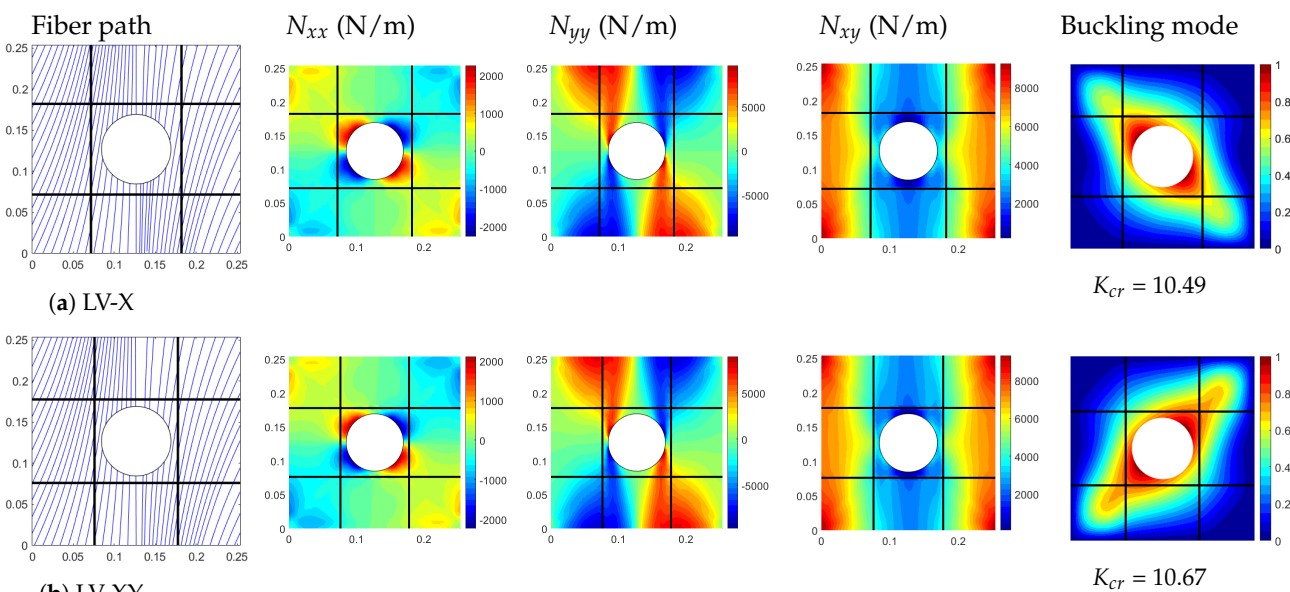

**Figure 17.** Comparison of optimal fiber paths, stress resultants, and buckling modes for different study cases of stiffened plate under in-plane shear displacements considering AFP manufacturing constraints. (**a**) LV-X laminate design; (**b**) LV-XY laminate design.

### 5.3. Load Case C: In-Plane Bending Displacement

For the panel subjected to in-plane pure bending displacements as seen in Figure 18, the in-plane boundary conditions are defined as:

- $x = 0$: $u = \dfrac{\Delta x}{(b/2)}(y - b/2)$ and $v = 0$;

- $x = a$: $u = -\dfrac{\Delta x}{(b/2)}(y - b/2)$ and $v = 0$;

- $y = 0$: $v = 0$;

- $y = b$: $v = 0$.

where $\Delta x = 0.01$ mm. During the buckling analysis, the plate's four edges are simply supported, $u = v = w = 0$.

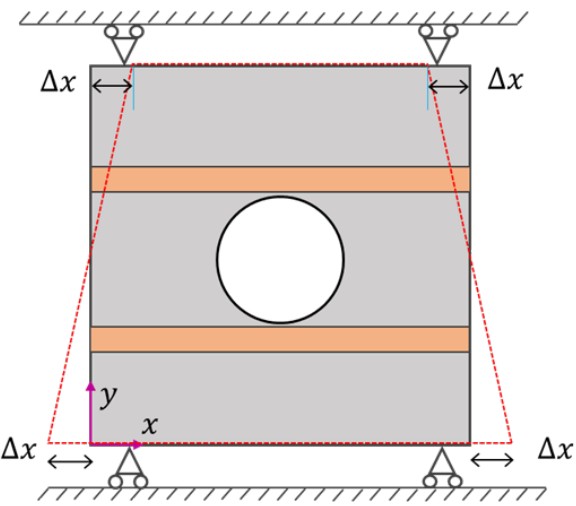

**Figure 18.** Composite stiffened plate under pure in-plane bending displacements.

### 5.3.1. Verification Studies

The stiffened plate model and the laminate configurations used in this study are same as those studied in Section 5.1.1. The applied in-plane bending displacements are shown in

Figure 18. The buckling mode results computed using the present program are in a good agreement with NASTRAN results, as seen in Figure 19. In the buckling mode results, due to the applied symmetric bending displacement about $x = a/2$, there are many pair modes each of which have the same values of the buckling load factors. Only one of each pair modes is selected for program validation in Figure 19. The stress resultants for the stiffened plate under in-plane pure bending displacements are computed and compared against the NASTRAN results. The two sets of results are found in a good agreement, as seen in Figure 20. The stiffener dominant axial stress distribution is also close to NASTRAN results, which are shown in Figure A1c in Appendix A for brevity.

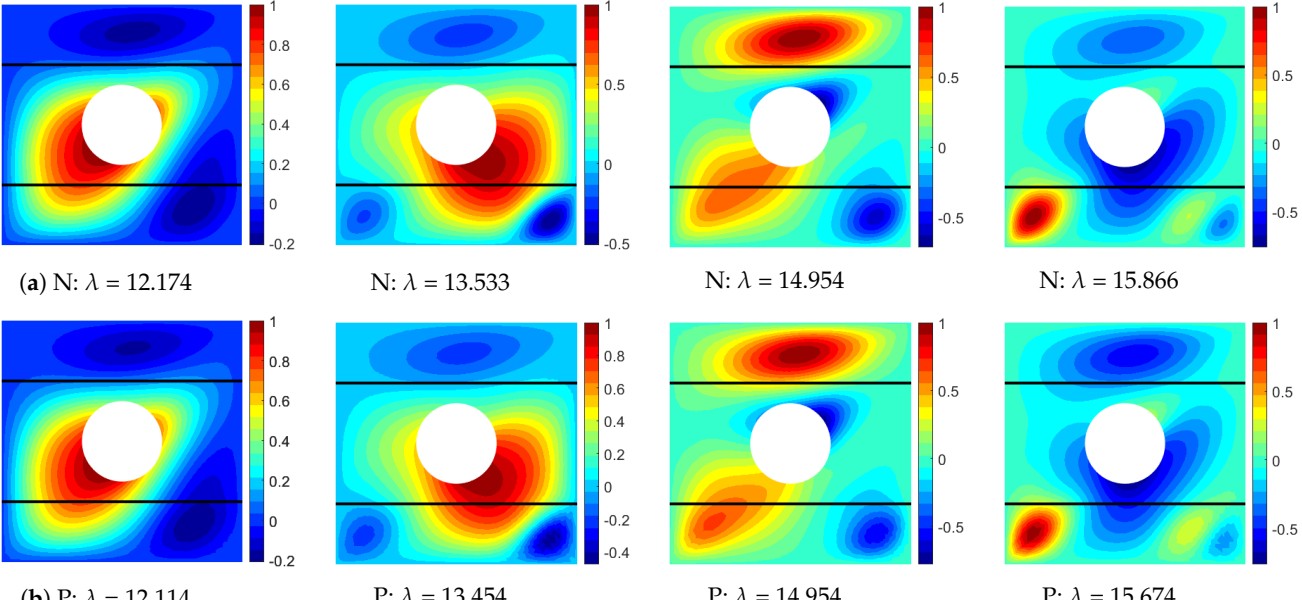

(**a**) N: $\lambda = 12.174$     N: $\lambda = 13.533$     N: $\lambda = 14.954$     N: $\lambda = 15.866$

(**b**) P: $\lambda = 12.114$     P: $\lambda = 13.454$     P: $\lambda = 14.954$     P: $\lambda = 15.674$

**Figure 19.** Comparison of buckling eigenvalue ($\lambda$) and buckling mode shapes of the stiffened plate under pure in-plane bending displacements. (**a**) N: NASTRAN results; (**b**) P: Present results.

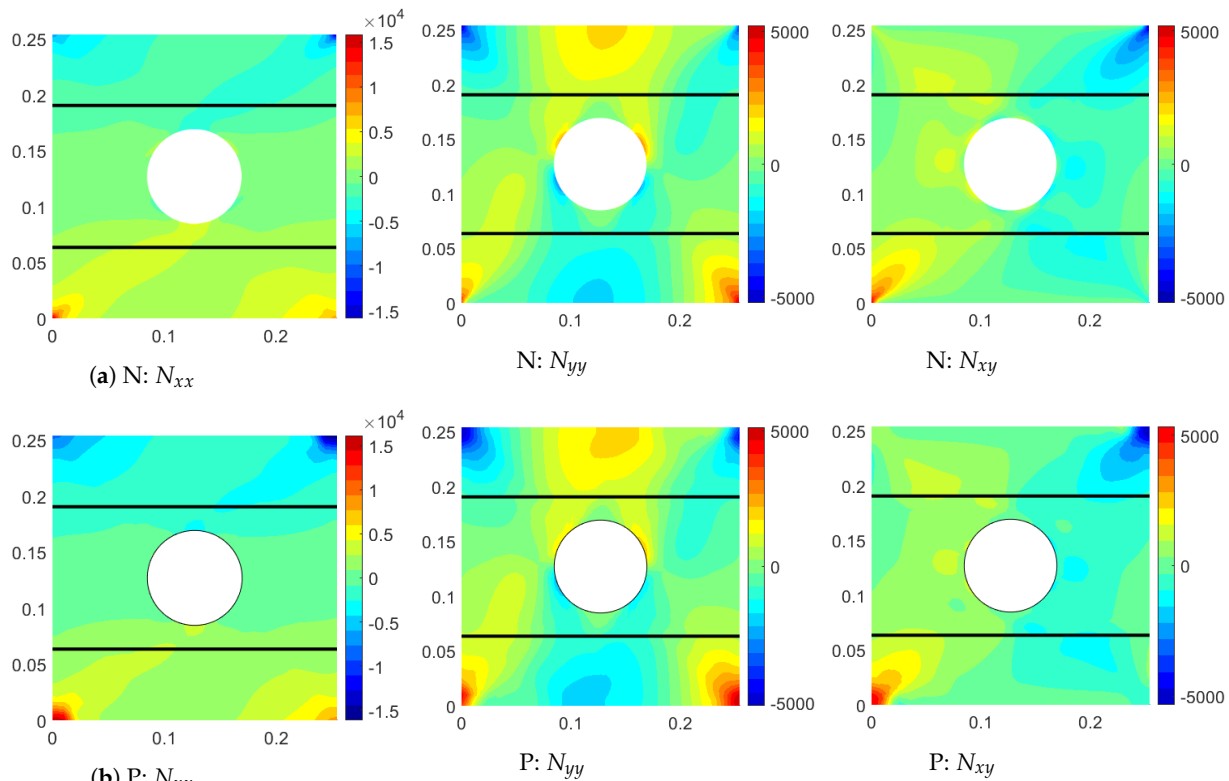

**Figure 20.** Comparison of stress resultants (unit: N/m) of the stiffened plate under pure in-plane bending displacements. (**a**) N: NASTRAN results; (**b**) P: Present results.

### 5.3.2. Optimization without Manufacturing Constraints

The optimal stiffener locations and fiber path orientations for different study cases of the stiffened plate with a central cutout under in-plane pure bending displacements are summarized in Table 8. The fiber paths, in-plane stress resultants, and the buckling mode results for different cases are shown in Figure 21. Table 8 shows that when using varying fiber path angle laminates for the plate, there are large increases in the buckling load, as much as 62.09% and 26.64%, respectively, as compared to those of the quasi-isotropic laminates and the optimal straight fiber path laminates.

**Table 8.** Optimal designs and results of the stiffened plate with a central cutout subjected to pure in-plane bending displacements (w/o AFP manufacturing constraints).

| Case | $\Theta$ | $\bar{x}$ | $K_{cr}$ | Improvement [1] | Improvement [2] |
|------|----------|-----------|----------|-----------------|-----------------|
| QI | $[(\pm 45/0/90)_2]_s$ | 0.319 | 3.93 | – | – |
| SF | $[(\pm 47.14)_4]_s$ | 0.000 | 5.03 | +28.00% | – |
| LV-X | $[(\pm \langle 40.78\|60.88\rangle)_4]_s$ | 0.000 | 5.21 | +32.58% | +3.58% |
| LV-XY | $[(90 \pm \langle -77.52\|76.15\rangle)_4]_s$ | 0.051 | 6.37 | +62.09% | +26.64% |

[1]: Compared to QI laminates results; [2]: compared to SF laminates result.

As expected, when using varying fiber path angle laminates with a rigid rotation to the whole laminates (LV-XY case), the locations for the maximum values for all three stress resultant components are shifted from the central hole's edge to the plate support edges, $y = 0$ and $y = b$ (see Figure 21d). Figure 21 also shows that for the laminate configurations of SF and LV-X cases, the optimal stiffeners are placed along the edges of the plate, $y = 0$ and $y = b$ where the large displacement magnitude is applied. The buckling modes for these two laminate configurations of SF and LV-X cases are almost identical. For the LV-XY design case, the optimal stiffener locations are also near the plate's edges $y = 0$ and $y = b$. This means that the stiffeners are mainly used to reduce the stress resultants for increasing

the buckling load for the stiffened VAT laminated plate with a central hole under pure in-plane bending displacements.

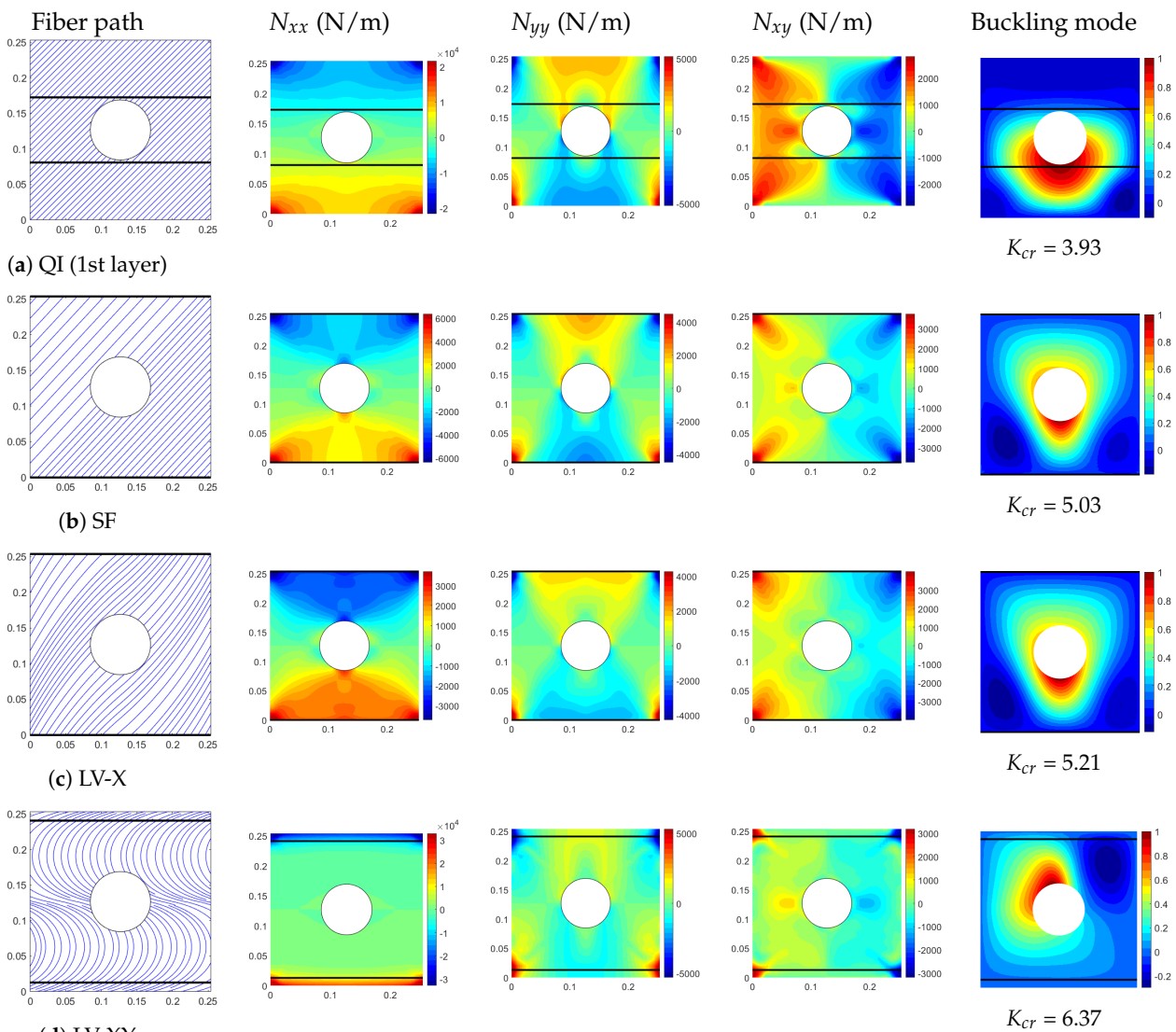

**Figure 21.** Comparison of optimal fiber paths, stress resultants, and buckling modes for different laminate configuration cases of the stiffened VAT laminates with a central hole under pure in-plane bending displacements. (**a**) QI laminate design; (**b**) SF laminate design; (**c**) LV-X laminate design; (**d**) LV-XY laminate design.

5.3.3. Optimization with Manufacturing Constraints

The AFP manufacturing constraints influence buckling response improvement of using VAT laminates for stiffened VAT laminated plate with a central hole under pure in-plane pure bending displacements. In this loading condition, the optimal fiber path orientations for the LV-XY design case are almost same with those for the LV-X design case, both cases increase buckling load to by 32.2% compared to the quasi-isotropic laminate cases (see Table 9). There is, however, a mere 3.4% increase in the buckling load as compared to the optimal straight fiber path laminate design results. The fiber path radius of curvature constraint is active in this optimization study, which causes a larger impact on the LV-XY case for a further buckling load increase, resulting in the buckling load to be 23.2–29.8% less than the one without considering AFP manufacturing constraints. The fiber paths, in-plane stress results and buckling mode results for different study cases are shown in Figure 22.

The optimal stiffeners are placed in the two edges, $y = 0$ and $y = b$ for all cases, which are used to reduce the in-plane axial stress resultant for improving the buckling responses.

**Table 9.** Optimal designs and results of the stiffened plate with a central cutout subjected to pure in-plane bending displacements (w/ AFP manufacturing constraints).

| Case | $\Theta$ | $\bar{x}$ | $K_{cr}$ | Improvement [1] | Improvement [2] |
|------|----------|-----------|----------|-----------------|-----------------|
| QI | $[(\pm 45/0/90)_2]_s$ | 0.319 | 3.930 | – | – |
| SF | $[(\pm 47.14)_4]_s$ | 0.000 | 5.025 | +28.00% | – |
| LV-X | $[(\pm\langle 42.26\vert 57.74\rangle)_4]_s$ | 0.000 | 5.196 | +32.21% | +3.40% |
| LV-XY | $[(0.07 \pm \langle 42.26\vert 57.74\rangle)_4]_s$ | 0.000 | 5.196 | +32.21% | +3.40% |

[1]: Compared to QI laminates results; [2]: compared to SF laminates results.

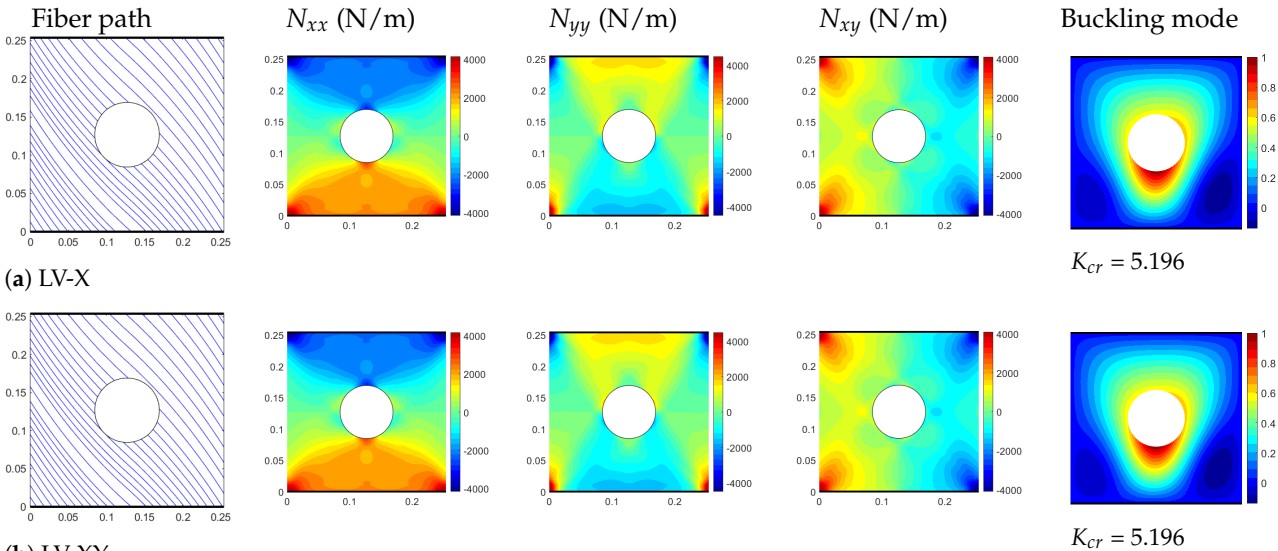

**Figure 22.** Comparison of optimal fiber paths, stress resultants, and buckling modes for different study cases of the stiffened VAT laminates with a central hole under in-plane bending displacements considering manufacturing constraints. (**a**) LV-X laminate design; (**b**) LV-XY laminate design.

## 6. Conclusions

This work presents buckling analysis and optimization of a stiffened, variable angle tow laminated plate with a central cutout under three different in-plane displacements. These three different in-plane displacements are three representative boundary conditions for aircraft semimonocoque components. Two common manufacturing constraints are considered: automatic fiber placement (AFP) head turning radius and the tow gap/overlap. These two manufacturing constraints are modeled by controlling the fiber path radius of curvature and adjacent tapes parallelism. To avoid a repeated meshing of the stiffened VAT laminated plate with a central cutout and to avoid a repeated checking of the mesh convergence during a stiffener shape optimization study, a previously developed modeling approach is employed to model the VAT laminates and stiffeners independently. The displacement compatibility is enforced at the stiffener–plate interface. The present modeling approach is verified by comparing the buckling mode results and stress resultants for a stiffened VAT laminated plate with a central hole under different boundary conditions against NASTRAN results. Buckling load maximization using the proposed method and a particle swarm optimization is performed. Some conclusions can be drawn from the present optimization studies:

- When there are no AFP manufacturing constraints considered in the design, the VAT laminates are highly effective in increasing the buckling load of the model under pure in-plane bending displacements, by up to 62.1% and 26.6%, respectively, as compared

to the quasi-isotropic and the optimal straight fiber path laminates, and the buckling load is increased by 21.2% and 12.4%, respectively, for the in-plane axial displacement case, and 19.7% and 12.5%, respectively, for the in-plane shear displacement case.

- When considering the AFP manufacturing constraints, the fiber path curvature constraint is found to be active for all design cases, and VAT laminates are mostly effective in increasing the buckling load of the model under in-plane shear displacements, by up to 16.4% and 9.4%, respectively, as compared to quasi-isotropic and optimal straight fiber path laminates, 11.1% and 3.0%, respectively, for the in-plane axial displacement case, and 32.2% and 3.4%, respectively, for the in-plane pure bending displacement case.
- The AFP manufacturing constraints restrain a large buckling load increase from using the VAT laminates for the model under in-plane axial and bending displacements, resulting in the maximum buckling loads 9.3–10.1% and 23.2–29.8% less than those obtained without considering AFP manufacturing constraints. They appear to have smaller impacts on the model under in-plane shear displacements, resulting in around 3% reduction in the maximum buckling load.
- The present study results provide some guidance in choosing the VAT laminates for different components as used in the aircraft design under different dominant loading conditions when considering AFP manufacturing constraints.

**Author Contributions:** Conceptualization, W.Z. and R.K.K.; methodology, W.Z. and R.K.K.; software, W.Z.; validation, W.Z.; formal analysis, W.Z.; investigation, W.Z. and R.K.K.; writing—original draft preparation, W.Z.; writing—review and editing, W.Z. and R.K.K. All authors have read and agreed to the published version of the manuscript.

**Funding:** This research received no external funding.

**Institutional Review Board Statement:** Not applicable.

**Informed Consent Statement:** Not applicable.

**Data Availability Statement:** NASTRAN input files and results presented in Sections 5.1.1, 5.2.1, and 5.3.1 for program validations, and MATLAB postprocessing subroutines have been stored in https://github.com/zhaowei0566/SPAD/tree/master/JournalofCompositeScience (accessed on 29 December 2021).

**Acknowledgments:** Wei Zhao thanks the technical support and high-performance computational resources provided by the Oklahoma State University High Performance Computing Center.

**Conflicts of Interest:** The authors declare no conflict of interest.

## Abbreviations

The following abbreviations, symbols, and Greek letters are used in this manuscript:

| | |
|---|---|
| $d$ | Width of the plate, m |
| $d_f$ | Width of the tape, mm |
| $\theta$ | Fiber path orientation, degrees |
| $\phi$ | Rigid rotation of lamina, degrees |
| $\Theta$ | A symbol used to describe the VAT lamina configuration, degrees |
| $\kappa$ | Fiber path curvature, 1/m |
| $\lambda cr$ | Buckling eigenvalue |
| $\Psi p$ | Buckling eigenvector |
| $r_{min}$ | Minimal radius of curvature of fiber path, m |
| $\vec{u}, \vec{v}$ | In-plane normal vectors used in the tape parallelism computation |
| $r$ | Radius of central circle cutout, m |
| $r_{min}$ | Minimal AFP head turning radius, m |
| $\vec{t}$ | Unit tangential vector of fiber path |
| $\Delta\alpha$ | Fiber path angle difference at the interface between two tapes, degrees |
| $\Delta x, \Delta y$ | Applied displacement along the $x$-axis and $y$-axis, m |

| | |
|---|---|
| $d_p$ | Plate element nodal displacement field vector, m |
| $\Delta d$ | Applied plate element nodal displacement, m |
| $K$ | Total stiffness matrix of the stiffened plate model |
| $K_p$, $K_s$, $K_{Gp}$ and $K_{Gs}$ | Plate elastic stiffness matrix, stiffener elastic stiffness matrix, plate geometric stiffness matrix, and stiffener geometric stiffness matrix |
| AFP | Auto fiber placement |
| LV-X | Linearly varying fiber path angle along the *x*-axis |
| LV-XY | Linearly varying fiber path angle along the *x*-axis and a rigid rotation of the VAT laminates |
| QI | Quasi-isotropic laminates |
| SF | Straight fiber path laminates |
| VAT | Variable angle tow |

## Appendix A. Stiffener Axial Stress Comparison

Figure A1 shows the comparison of stiffener axial stress distribution for the verification examples studied in Sections 5.1.1, 5.2.1, and 5.3.1. For the in-plane shear displacement boundary condition case, the stiffeners located at $x = a/4$ and $y = b/4$ have the same axial stress distribution, and the stiffeners located at $x = 3a/4$ and $y = 3b/4$ have the same axial stress distribution. The first node of each stiffener beam element is chosen where the axial stress is computed and compared.

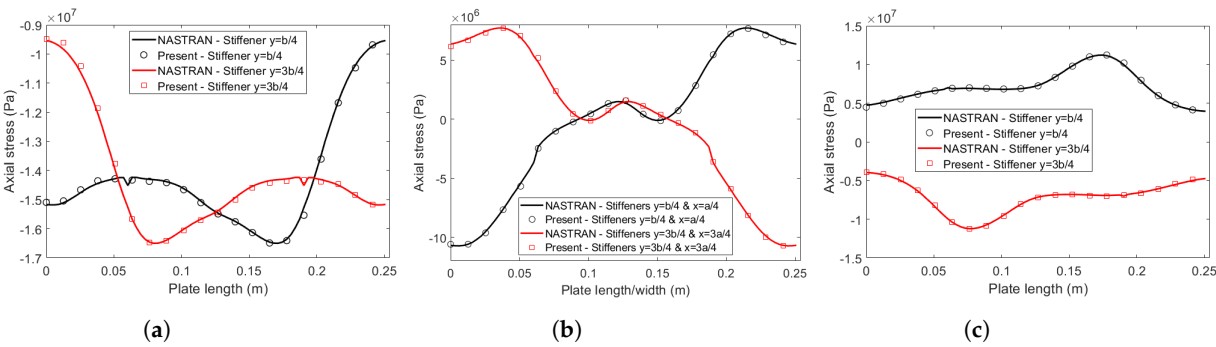

(**a**)          (**b**)          (**c**)

**Figure A1.** Comparison of axial stress (unit: N/m$^2$ or Pa) of the stiffeners under different loading conditions. (**a**) In-plane axial displacement; (**b**) in-plane shear displacement; (**c**) in-plane bending displacement.

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
