# Peer review of "Buckling Analysis and Optimization of Stiffened Variable Angle Tow Laminates with a Cutout Considering Manufacturing Constraints"

_jcs, doi:10.3390/jcs6030080_

Round 1
Reviewer 1 Report
Peer review Report on:
“Buckling analysis and optimisation of stiffened variable angle tow laminates with cutouts considering manufacturing constrains”
The proposed manuscript is of great interest with no major issues to be addressed; based on a stiffened variable angle tow laminated plate 3D model, a buckling load maximization study is offered from manufacturing head turning radius and tow overlap/gaps point of view. Three main loading (boundary) conditions were considered: in-plane axial displacement (fuselage panels case), in-plane pure shear displacement (wing skin panels case) and in-plane pure bending displacement (rib panels, spar panels case).
The buckling load optimization is realised by means of maximization of the normalized buckling parameter for each load (boundary) condition. Because of non-convex fitness function in terms of fibre path orientation, a particle swarm optimisation (PSO) has been conducted.
We would suggest the replacement of “cutouts” with “cut-outs”, “fibers” with “fibres”;
A better representation (scaling if possible) concerning Figure 1 (b) – in order to emphasize the difference between fibres with dissimilar path angle;
We believe that Figure 5 depicts a stiffened plate with a central cut-out (not without);
To be checked in Figure 9 comment, stress measurement unit should be N/mm2 (the same for Figure 15, 20);
Should replace “good agreement” with “convergent result array”, line 363;
If possible, one could use an example for the stiffener axial stress distribution comparison, line 366.
Author Response
Re: Response Letter for Manuscript ID jcs-1601019
The proposed manuscript is of great interest with no major issues to be addressed; based on a stiffened variable angle tow laminated plate 3D model, a buckling load maximization study is offered from manufacturing head turning radius and tow overlap/gaps point of view. Three main loading (boundary) conditions were considered: in-plane axial displacement (fuselage panels case), in-plane pure shear displacement (wing skin panels case) and in-plane pure bending displacement (rib panels, spar panels case).
- The buckling load optimization is realised by means of maximization of the normalized buckling parameter for each load (boundary) condition. Because of non-convex fitness function in terms of fibre path orientation, a particle swarm optimisation (PSO) has been conducted.
Response: We thank the reviewer for reviewing this manuscript. We have revised the manuscript based on your comment.
- We would suggest the replacement of “cutouts” with “cut-outs”, “fibers” with “fibres”;
Response: We have revised them as suggested.
- A better representation (scaling if possible) concerning Figure 1 (b) – in order to emphasize the difference between fibres with dissimilar path angle;
Response: This figure has been updated.
- We believe that Figure 5 depicts a stiffened plate with a central cut-out (not without);
Response: We apologize for this typo. It has been corrected.
- To be checked in Figure 9 comment, stress measurement unit should be N/mm2 (the same for Figure 15, 20);
Response: They are stress resultants, which are computed as where is measured about the plate’s reference plate. is the number of laminas. Therefore, the unit is N/m. The stress resultant was used to compute the geometric stiffness for the subsequent buckling eigenvalue computation.
- Should replace “good agreement” with “convergent result array”, line 363;
Response: We are not clear what is the “array” as suggested by the reviewer.
- If possible, one could use an example for the stiffener axial stress distribution comparison, line 366.
Response: The comparisons of the stiffener axial stress distribution have been studied for all three loading conditions. To save space, the comparison figures have been moved to Appendix. Please see Figure 1 in Appendix A of the revised manuscript.
Reviewer 2 Report
Thank you for submitting your paper. The work done here draws attention to a significant subject in buckling of two laminates. I have found the paper to be interesting. However, several issues need to be addressed properly before the paper is being considered for publication. My comments including major and minor concerns are given below:
- Please consider reviewing the abstract and highlight the novelty, major findings, and conclusions. I suggest reorganizing the abstract, highlighting the novelties introduced. The abstract should contain answers to the following questions:
- What problem was studied and why is it important?
- What methods were used?
- What conclusions can be drawn from the results? (Please provide specific results and not generic ones).
- The abstract must be improved. It does not read well at all. Please use numbers or % terms to clearly shows us the results in your experimental work. Please expand the abstract.
- Please consider reporting on studies related to your work from mdpi journals.
- The introduction must be expanded, please consider improving the introduction, provide more in-depth critical review about past studies similar to your work, mention what they did and what were their main findings then highlight how does your current study brings new difference to the field.
- The authors need to add a list of nomenclature for all the abbreviations, symbols, and Greek letters at the end of the manuscript.
- Page 7-10 this reads more like a student thesis or a set of instructions for a setup, it is too long and is not suitable for a scientific article, I strongly recommend the authors to shorten it and keep the main parts only. Reduce it to 1 or 1.5 pages and the rest can either be moved to an appendix or removed.
- Table 1 needs a reference is not measured by the authors.
- Buckling Analysis Governing Equations again too long, seems like it was cut paste from a thesis, please shorten as possible.
- Line 257 “which will not be shown here in detailed.” Changed detailed to details.
- Please shorten pages 11-15 into 2 pages or less. Everything else can to an appendix
- Line 251 the authors are stating obvious facts, when I read this paragraph its like reading a manual trying to teach basics of meshing in FE software. Please avoid any unnecessary or excessive information which does not add value to the paper.
- Line 298 remove the word “very”
- Line 300 please rephrase it does not read well.
- Moderate English correction is required.
- Table 3: Summary of Design Variables. Why only two levels each? They are quiet few, is that enough for the optimisation study?
- The results are merely described and is limited to comparing the experimental observation and describing results. The authors are encouraged to include a more detailed results and discussion section and critically discuss the observations from this investigation with existing literature.
- Conclusion can be expanded or perhaps consider using bullet points (1-2 bullet points) from each of the subsections.
- Please shorten and reorganise the thoughts/flow of information in the article. Provide clear sections and subsections. remove all uncessary information and move some results to an appendix, it is difficult to track all the work done and the overall purpose of this work. It looks more like several optimisation attempts with different setup conditions.
- Overall paper is worthy of publication after revising it.
Round 2
Reviewer 2 Report
All questions answered and paper can be accepted